# Design and Dynamic Stability Analysis of a Submersible Ocean Current Generator Platform Mooring System under Typhoon Irregular Wave

Shueei-Muh Lin [1,2,]*, Chihng-Tsung Liauh [1,2] and Didi-Widya Utama [2]

[1] Green Energy Technology Research Centre (GETRC), Kun Shan University, Tainan 71070, Taiwan; liauhct@mail.ksu.edu.tw
[2] Department of Mechanical Engineering, Kun Shan University, Tainan 710, Taiwan; didiu@ft.untar.ac.id
* Correspondence: smlin45@gmail.com

**Abstract:** This research proposes a mooring system for an ocean current generator that is working under the impact of typhoon waves. The turbine and the platform are kept stable at a designed water depth to ensure that the generator remains undamaged and continuously generates electricity under excessive water pressure. In this design, the turbine generator is mounted in front of the floating platform by ropes and withstands the force of ocean currents, while the platform is anchored to the deep seabed with lightweight, high-strength PE ropes. In addition, two pontoons are used to connect the generator and the platform with ropes. When the balance is reached, the depth of the generator and the depth of the platform's dive can be determined by the length of the ropes. In this study, typhoon irregular wave is represented by the Jonswap wave spectrum. The irregular wave is simulated by six regular waves. The equation of motion of the mooring system is derived. The theoretical solution of the dynamic system is presented to determine the dynamic displacements of the platform, pontoon, turbine and the dynamic tensions of the ropes. The dynamic tensions of the ropes increase with the cross-sectional area of pontoon. The natural frequency of the mooring system depends on the parameters, including the mases of elements, the lengths of ropes and the cross-sectional area of pontoons. In the proposed mooring configuration, the dynamic tension of the rope is far less than the breaking strength of the rope; thus, the ocean turbine is stable, and no water that flows through will be disturbed by the floating platform.

**Keywords:** dynamic tension; displacement; ocean current; floating platform; turbine; pontoon; buffer spring

## 1. Introduction

An excellent, natural energy resource is the Kuroshio strong current flowing along the east of Taiwan, which has an estimated electricity capacity of 4 GW [1]. Ocean current is one of the potential energy sources to be developed. However, the seabed beneath the Kuroshio current is almost over 1000 m in the area mentioned above. Moreover, several typhoons strike Taiwan every year. These two disadvantages must be solved before a current power generation system is constructed.

Investigation of fluid–structure interaction (FSI) is important for marine engineering, aircraft, engines, bridges and biotechnology. FSI is the interaction of some movable or deformable structures with an internal or surrounding fluid flow. Fluid–structure coupling can be simply divided into one-way coupling and two-way coupling. One-way coupling ignores the change of flow field space caused by structural deformation, so the calculation is more simplified. Anagnostopoulos [2] investigated the dynamic response analyses of offshore platforms under wave loadings and predicted the wave forces by means of Morison's equation. Therefore, the equation of motion for the lumped mass idealization of the platform was presented. The system was one-way coupling. It was found that the

importance of fluid–structure interaction increased with higher dynamic amplifications. The effect of viscous damping due to the relative velocity between fluid and structure significantly decreased the resonant response. Istrati and Buckle [3] investigated the effect of FSI on connection forces in bridges caused by tsunami loads by using LS-DYNA software. It was found that the flexibility and the dynamic characteristics of the bridge structure significantly influenced the external tsunami loads on the bridge and the connection forces.

Xiang and Istrati [4] investigated the solitary wave–structure interaction of complex coastal deck geometries by using the Lagrangian–Eulerian (ALE) method with a multi-phase compressible formulation. It was found that, for small wave heights, the horizontal and uplift forces increased with the number of girders (Ng), while, for large waves, the opposite happened. Moreover, if the Ng was small, the wave particles accelerated after the initial impact on the offshore girder, leading to more violent slamming and larger pressures and forces on the deck. Conversely, if the Ng was large, unsynchronized eddies were formed in each chamber, which dissipated energy and resulted in weaker impacts on the deck. Obviously, if the surfaced structure is too large, the two-way coupling effect of wave–structure interaction needs to be considered. In addition, the multi-phase flow simulation needs to be considered in the numerical analysis, which is a very challenging problem and important in marine engineering. Some literature [4–6] is devoted to this research. Peregrine et al. [5] found that the breaking/broken waves and bores were dominated by significant turbulence effects and air entrapment. The hydrodynamic loads caused by the breaking wave on the marine decks were totally different from unbroken waves [6].

Firouz-Abadi et al. [7] investigated the stability analysis of shells conveying fluid. The boundary element method was applied to model the potential flow. It was found that the eigenvalues and mode shapes of the flow in the shell were strongly related to the unsteady pressure that induced the shell vibration. Bose et al. [8] investigated the flow-induced dynamic stability of a fluid–structure interaction (FSI) system comprising of a symmetrical NACA 0012 airfoil supported by non-linear springs. Lin et al. [9] investigated the wave propagation of an artery. A mathematical model was proposed to describe the wave propagation through an isotropic, elastic, thick tube filled with viscous and incompressible fluid. The tube is supported by the elastic muscle and simulated as the viscoelastic foundation. The flexural Young and Lamb wave modes through a tube wall are presented simultaneously. The dispersion curves and the energy transmissions of the three modes were investigated. It was found that the effect of the viscoelastic foundation constant on the wave speed and the transmission was significant.

The numerical method is usually used to investigate the dynamic behavior of the two-way-coupled FSI. In general, the numerical methods include the boundary element method [10], the finite volume method [11], the finite-element-based, arbitrary Lagrangian–Eulerian method [12], particle-based methods, such as smoothed particle hydrodynamics [13], and hybrid methods, such as coupled SPH-DEM [14] and coupled SPH-FEM [15].

There have been two cases where the performance of ocean current turbines was tested in seas: (1) One 50 kW ocean current turbine, developed by the Wanchi company (Kaohsiung City, Taiwan), was successfully moored to the 850 m deep seabed near the offshore of Pingtung County, Taiwan, by Chen et al. [1]. The current turbine generated about 26 kW under the current speed of 1.0 m/s; (2) An experimental 100-kW-class ocean current turbine was located off the coast of Kuchinoshima Island, Kagoshima Prefecture, and demonstrated by IHI and NEDO [16]. The current turbine generated about 30 kW under the current speed of 1.0 m/s. The turbine system 50 m below the sea surface was connected to the mooring foundation on the seabed at the depth of 100 m. The above experiments were conducted under the condition of small waves, and the influence of waves on the dynamic stability of the mooring system was not studied.

Zwieten et al. [17] investigated the C-plane prototype of an ocean current turbine with a hydrodynamic platform that was connected to the seafloor with a rope. This turbine, using its wingtips and canard to manipulate its depth and orientation in a temporally and spatially varying current, could generate maximum energy production. This study did not

take the problem of turbine damage due to excessive water pressure when diving too deep into consideration. It also did not consider the disadvantages of the deeper ocean current, the lower flow rate and the smaller power generation. The effect of waves was also not considered.

One of the most challenging tasks for the ocean current turbine system is to develop a deep mooring technology because the targeted seabed is at a depth of almost 1000 m, as mentioned. To monitor the performance of the ocean turbine, the dynamic stability of the mooring system under the coupled effect of the ocean current and wave is needed [18–24]. Lin et al. [25] used the ocean current turbine system developed by the Wanchi company to investigate the dynamic stability of the system subjected to regular wave and current forces. The mooring system was composed of a turbine, a floating platform, traction ropes and a mooring foundation. Results showed that the effects of several parameters of the system on the dynamical stability of the ocean current turbine system were significant. However, the dynamic tension of the rope was not investigated in the study.

As the mooring foundation is set on the seabed over 1000 m deep, a long mooring rope is required. Consider the strength of the rope: lightweight, high-strength PE mooring ropes are more beneficial than chain and steel ropes. Lin and Chen [26] found that, when the ocean current velocity was 1 m/s and the rope length was about 2900 m, the drag force was 15 tons, and the rope was almost straight. In other words, the bending deformation of the PE rope was negligible. The deformation of the rope was longitudinal only. Accordingly, the mooring system is simulated in the linear elastic model to analyze the problem of dynamic stability. Consider an ocean current power generation system composed of a surfaced turbine, a floating platform, a towing rope and a mooring foundation [25]: whenever a typhoon hits, the turbine generator is towed back to the shore to avoid any possible damage, leaving the mooring system in the sea. Lin and Chen [26] proposed a protection method to protect the mooring system that avoids the damage caused by typhoon wave current. The principle of the design is that the platform generates a negative buoyancy to dive by letting water flow into its inner tank, and the pontoon is used to create a positive buoyancy. When the two elements are connected by a rope to achieve static equilibrium, the floating platform is submerged at a fixed depth determined by the rope length. Furthermore, the linear elastic model is used to construct the coupled motion equation of the system under a regular wave. The analytical solutions of the coupled equations are derived. It is theoretically verified that the proposed protection procedure can avoid the damage of the floating platform and the mooring line due to typhoon wave impact.

Lin et al. [27] simulated a mooring system for ocean current generation during non-typhoon periods and proposed a system that keeps the turbine statically stable at a designed underwater depth to ensure that the ocean current generator can generate electricity effectively. In their design, the turbine generator is connected to a surfaced platform, the platform is anchored to the deep mooring foundation by lightweight, high-strength polyethylene ropes and a pontoon is connected to ocean current turbines with rope. The static balance of the ocean current turbine is formed. Therefore, the depth of the current turbine can be determined by the length of the rope. Additionally, the linear elastic model is used to simulate the motion equation of the overall mooring system under a regular wave. The theoretical solution of the static and dynamic stability analysis of the mooring system is proposed. The dynamic displacements of the components and the dynamic tensions of ropes under the regular wave and ocean current are investigated. It is found that the effect of the wave phase on the dynamic response of the system is significant. The length of the rope can be adjusted to avoid resonance and reduce the tension of the rope. In addition, a buffer spring is used to reduce the dynamic tension of the rope to increase the safety and lifespan of the rope significantly.

To simplify the actual ocean waves, which are irregular, three approaches are commonly adopted: (1) the approximation of the wave field by a single, sinusoidal component with a given height, period and direction (regular waves); (2) the use of a limited number of harmonics of a primary wave to approximate non-sinusoidal properties (irregular waves); and (3) the representation of the water surface by an infinite summation of Fourier

components (wave spectrum) [28]. Pierson and Moskowitz [29] presented the Pierson and Moskowitz wave spectrum. The assumption was that, if the wind blows steadily over a large area for a long time, the waves will reach equilibrium with the wind. This is the concept of a fully developed sea, which requires winds of a sea that continuously blow over hundreds of miles for several days to reach full development. Hasselmann et al. [30] experimentally found that the wave spectrum can never be fully developed. The wave spectrum continues to develop due to wave-to-wave interactions, even over long periods of time and distances. Therefore, the Pierson–Moskowitz spectrum is modified to add an additional and somewhat artificial factor to it to make the wave spectrum and experimental measurements more closely matched. The Jonswap wave spectrum is presented.

This study proposes a mooring design in which the ocean current generator still generates electricity when typhoon waves hit without interruption. To prevent the typhoon waves from invading the ocean current generator set, a process is adopted whereby the system dives below 60 m to avoid the damage of the typhoon waves. At the same time, to prevent diving too deep from damaging the turbine, the turbine is in a static balance at a predetermined depth underwater, and it must be able to maintain the a not-too-large dynamic displacement. The surface velocity of the Kuroshio in eastern Taiwan is relatively fast, and the deeper the water depth, the smaller the velocity. Therefore, the ocean current generator group should not be placed too deep. This study proposes a safe and efficient mooring system design and a linear elastic model to simulate the motion of the entire mooring system. Results for analyzing the static and dynamic stability of mooring systems, the dynamic displacements of turbines, floating platforms, pontoons and the dynamic tension of ropes under the action of typhoon waves and ocean currents are studied. The effects of several parameters on the dynamic behavior of the system are presented.

## 2. Mathematical Model

As shown in Figures 1 and 2, to prevent the damage of the typhoon waves, the turbine and the floating platform are submerged to a depth of less than 60 m. Therefore, the influence of the wave impact is almost negligible. In general, the dynamic response of a large-surfaced structure subjected to wave impact force, which is non-uniform and transient, is generally in the coupled translational–rotational (pitching, rolling and yawing) motion. The good conditions for ocean current power generation are high flow rate and stable flow direction, so the site is often a considerable distance from the shore: less affected by the coast and less likely to produce breaking waves. The impact of breaking waves is not considered in this manuscript.

When ocean currents flow through the blades of the ocean turbine, the turbine rotates and drives the power generator to generate electricity. Meanwhile, the turbine unit is subjected to the force of the ocean current; to fix the turbine unit, it is pulled by the floating platform connected by rope B. The floating platform provides buoyancy and is anchored to the deep seabed with lightweight, high-strength PE ropes. In addition, the ocean current turbine is connected to pontoon 4 via rope D, and the balance between the current generator and the pontoon is reached so that the depth of the turbine when the current is not affected can be determined by the length $L_C$ of rope C. On one side of the floating platform, rope B is used to pull the ocean current generator, and the other side of the floating platform is pulled down and anchored on the deep seabed. The buoyancy of the floating platform can be adjusted to be smaller than that of static balance so that, when ropes A and B are pulled, the floating platform has negative buoyancy and pontoon 3 has positive buoyancy, and rope C is used to connect the floating platform and pontoon 3 to achieve a balance of positive and negative buoyancy. In this way, the depth of the floating platform can be calculated by the length $L_C$ of rope C.

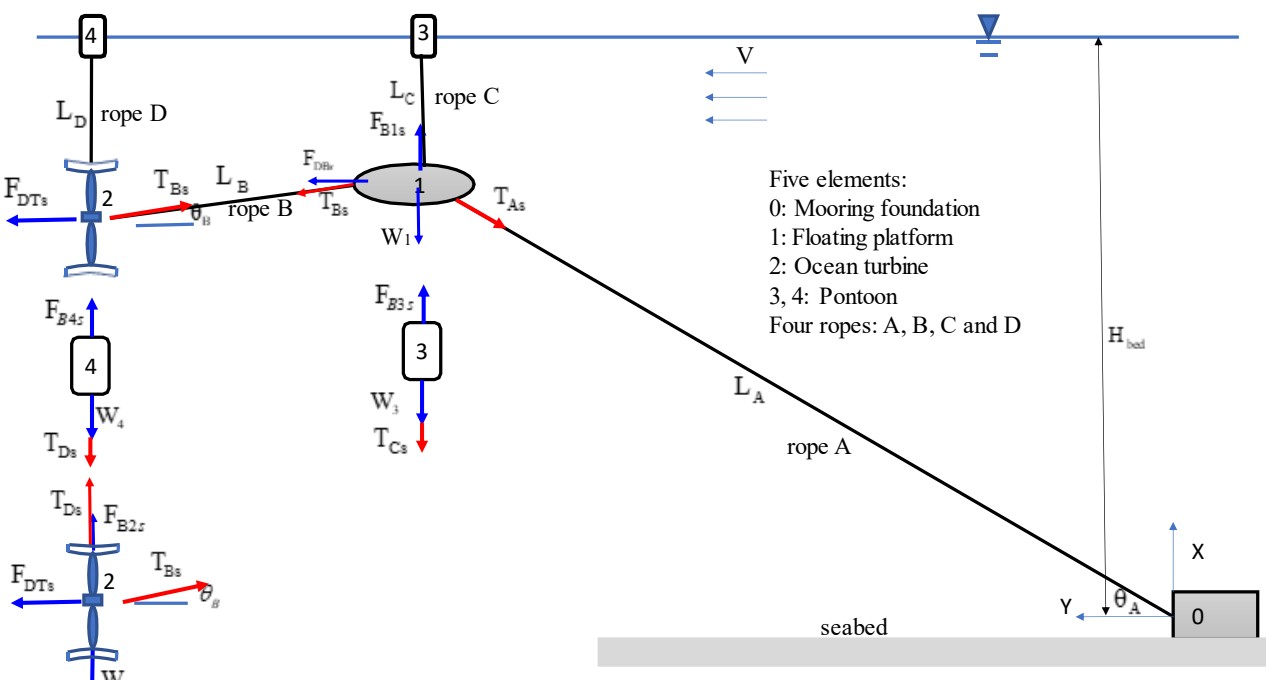

**Figure 1.** Coordinates of the current energy system composed of submarined ocean turbine, pontoons, floating platform, traction ropes and mooring foundation in the static state under steady ocean current.

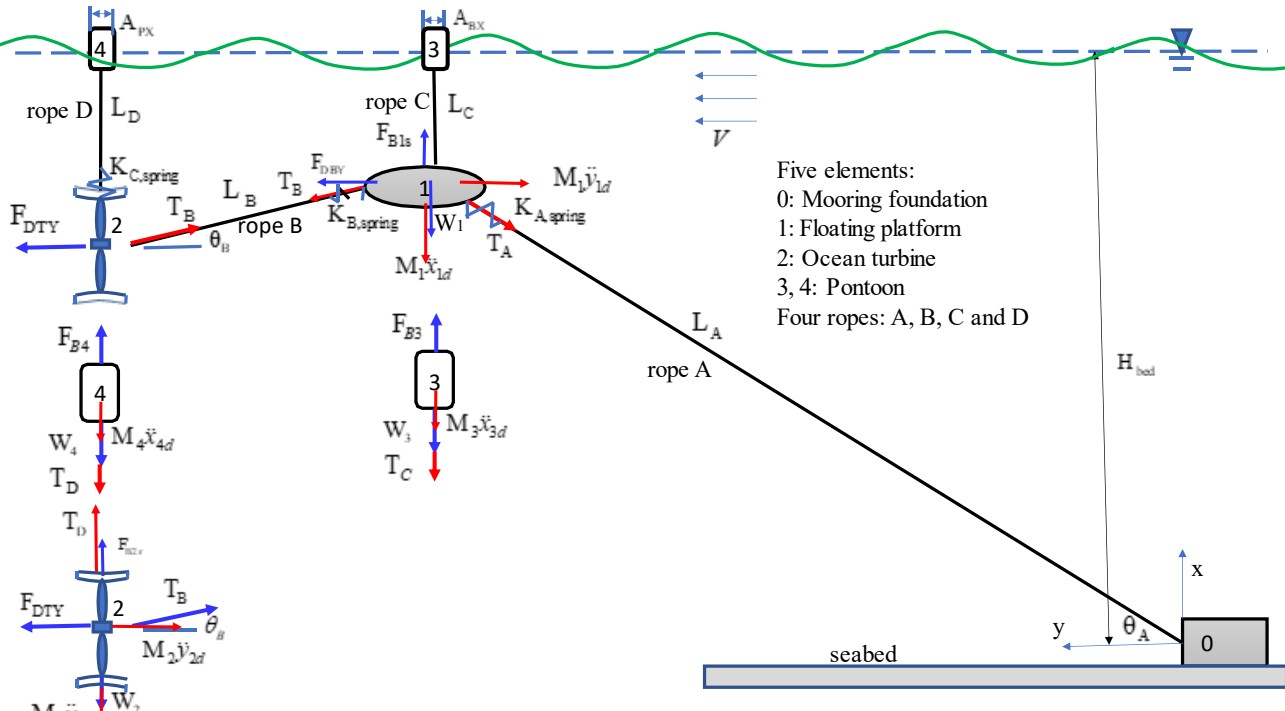

**Figure 2.** Coordinates of the current energy system composed of submarined ocean turbine, pontoon, floating platform, traction rope and mooring foundation in the dynamic state under steady ocean current and wave.

Lin and Chen [26] showed that a PE rope can be assumed to be a straight line under a certain amount of ocean current drag force because the force deformation of the PE rope is negligible. The linear elastic model presented by Lin and Chen [26] is used to analyze the motion equation of the overall mooring system.

Several assumptions are made based on these facts about ocean current energy converters (OCEC):

- The current flow is steady;
- The masses of the turbine, floating platform and the pontoon are concentrated;
- Lightweight and high-strength PE mooring ropes are used;
- Under the ocean velocity, the deformed configuration of PE rope is nearly straight;
- The elongation strain of the ropes is small;
- The tension of the rope is considered uniform.

According to these assumptions, the motion of the mooring system is translational. The coupled translational–rotational motion of a system subjected to non-uniform and impulsive force from the wave current will be discussed in future research. The coupled, linear, ordinary differential equations of the system are derived based on the assumptions. Due to the wave fluctuation, the buoyance forces of the pontoons stimulate the mooring system to vibrate. The coupled vibration motion of the system includes horizontal and vertical oscillations.

The global displacements $(x_i, y_i)$ for the $i$-th element shown in Figures 1 and 2 are the sum of two parts: (1) the static one subjected to the steady current and (2) the dynamic one subjected to the wave, as follows:

$$x_i = x_{is} + x_{id}, \quad y_i = y_{is} + y_{id}, \ i = 1, 2, 3, 4 \tag{1}$$

where $x$ and $y$ are the vertical and horizontal displacements, respectively. Because of the pontoon buoyancy and the short length of rope between the turbine and the pontoon, the horizontal dynamic displacements of the turbine and pontoon 4 are almost the same: $y_{2d} \approx y_{4d}$. In a similar way, the horizontal dynamic displacements of the floating platform and pontoon 3 are almost the same: $y_{1d} \approx y_{3d}$. In addition, the total tensions of ropes A, B, C and D are also composed of two parts: (1) the static one and (2) the dynamic one, as follows:

$$T_i = T_{is} + T_{id}, \ i = A, \ B, \ C, \ D \tag{2}$$

### 2.1. Static Displacements and Equilibrium under the Steady Current and without the Wave Effect

The static displacements of the five elements are:

$$
\begin{aligned}
x_0 &= 0, \ y_0 = 0 \\
x_{1s} &= H_{bed} - L_C = L_A \sin\theta_{As}, \ y_{1s} = L_A \cos\theta_{As}; \\
x_{2s} &= H_{bed} - L_D = x_{1s} - L_B \sin\theta_{Bs}, \ y_{2s} = y_{1s} + L_B \cos\theta_{Bs} \\
x_{3s} &= x_{1s} + L_C = H_{bed}, \ y_{3s} = y_{1s} \\
x_{4s} &= x_{3s} = x_{2s} + L_D = H_{bed}, \ y_{4s} = y_{2s}
\end{aligned}
\tag{3}
$$

Due to $x_{1s} >> x_{1d}$, the global inclined angle $q_A$ can be expressed as:

$$\sin\theta_A = \frac{x_1}{L_A} = \frac{x_{1s} + x_{1d}}{L_A} \approx \frac{x_{1s}}{L_A} = \sin\theta_{As} \tag{4}$$

Due to $x_{is} >> x_{id}$, the global inclined angle $q_B$ can be expressed as:

$$\sin\theta_B = \frac{x_1 - x_2}{L_B} = \frac{(x_{1s} + x_{1d}) - (x_{2s} + x_{2d})}{L_B} \approx \frac{x_{1s} - x_{2s}}{L_B} = \sin\theta_{Bs} \tag{5}$$

Under the steady current and without the wave effect, the static horizontal and vertical equilibriums of the floating platform are written, respectively, as shown in Figure 1.

$$T_{Bs} \cos\theta_{Bs} + F_{DBs} = T_{As} \cos\theta_{As} \tag{6}$$

$$F_{B1s} = T_{As} \sin\theta_{As} + T_{Bs} \sin\theta_{Bs} + W_1 \tag{7}$$

where $T_{As}$, $T_{Bs}$, $F_{B1s}$ and $W_1$ are the static tensions of ropes $A$ and $B$, the buoyancy of the floating platform and the weight of the floating platform, respectively. The steady drag of the floating platform under current $F_{DFs} = \frac{1}{2}C_{DFy}\rho A_{FY}V^2$.

The static horizontal and vertical equilibriums of the turbine are expressed, respectively, as:

$$T_{Bs}\cos\theta_{Bs} = F_{DTs} \tag{8}$$

where the steady drag of the turbine $F_{DTs} = C_{DTy}\frac{1}{2}\rho A_{Ty}V^2$.

$$F_{B2s} = W_2 - T_{Ds} - T_{Bs}\sin\theta_{Bs} \tag{9}$$

where $T_{Ds}$, $F_{B2s}$ and $W_2$ are the static tensions of rope D, the static buoyancy and the weight of the turbine, respectively. The static vertical equilibrium of pontoon 3 is expressed as:

$$F_{B3s} = W_3 + T_{Cs} \tag{10}$$

where $F_{B3s}$ and $W_3$ are the static buoyancy and the weight of pontoon 3, respectively. The static vertical equilibrium of the pontoon 4 is expressed as:

$$F_{B4s} = W_4 + T_{Ds} \tag{11}$$

where $F_{B4s}$ and $W_4$ are the static buoyancy and the weight of pontoon 4, respectively.

### 2.2. Simulation of Irregular Wave

The irregular wave is represented by the Jonswap wave spectrum. The Jonswap wave spectrum is given as a modification of the Pierson–Moskowitz spectrum in accordance with DNV [28,31].

The wave energy spectrum is:

$$S_J(f) = B_J H_s^2 f_p^4 f^{-5} \exp\left[\frac{-5}{4}\left(\frac{f}{f_p}\right)^{-4}\right]\gamma^b \tag{12}$$

where $f$ is the wave frequency, $f_p$ is the peak frequency and $H_s$ is the significant wave height.

$$B_J = \frac{0.06238\times(1.094-0.01915ln\gamma)}{0.230+0.0336\gamma-\frac{0.0185}{1.9+\gamma}}, \quad b = \exp\left[-0.5\left(\frac{f-f_p}{\sigma f_p}\right)^2\right],$$

$$\sigma = \left\{ \begin{array}{l} 0.07, \text{ for } f \le f_p \\ 0.09, \text{ for } f > f_p \end{array} \right. , \gamma = 3.3 \tag{13}$$

Referring to the information from the Central Meteorological Bureau Library of Taiwan about the typhoons that have invaded Taiwan from 1897 to 2019 [26,32] and selecting 150 typhoons that greatly affected Taiwan's Green Island, the significant wave height $H_s$ during the 50-year regression period $H_s$ =15.4 m, and the peak period $P_w$ = 16.5 s.

Substituting the significant wave height $H_s$ and the peak period $P_w$ into Equations (12) and (13), the Jonswap wave spectrum is determined, as shown in Figure 3.

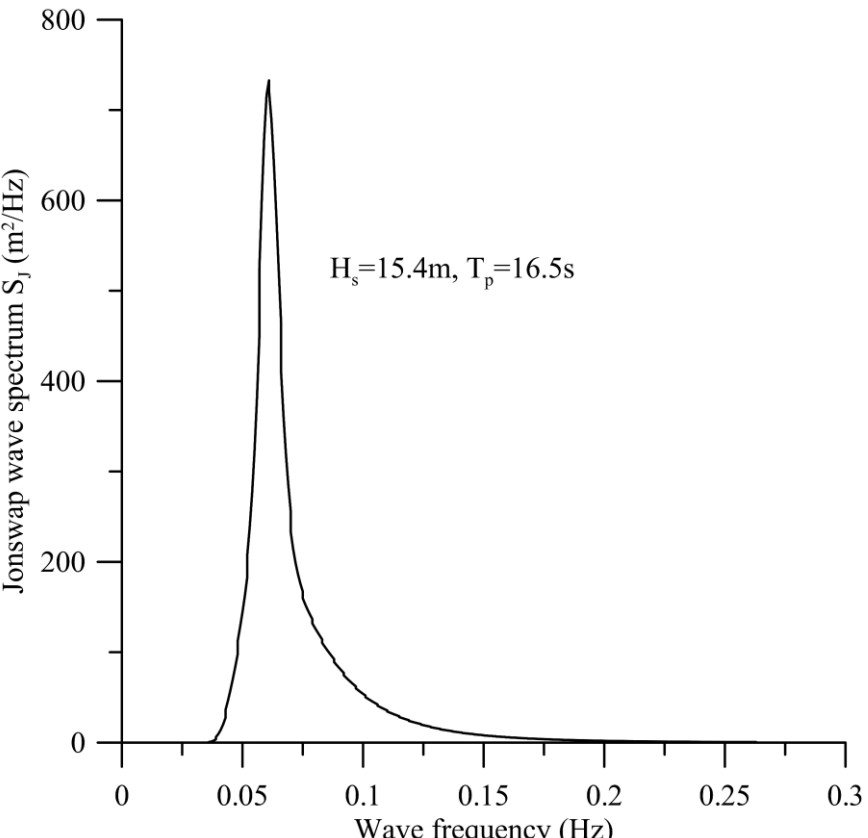

**Figure 3.** Jonswap wave spectrum.

The frequency domain of the wave spectrum is divided into $N$ subdomains of $(\omega_0, \omega_1), \dots, (\omega_{N-1}, \omega_N)$. The sea surface elevation of an irregular wave can be generated by the superposition of the regular wave components:

$$x_w = \sum_{i=1}^{N} a_i \sin\left( \Omega_i t - \vec{K}_i \cdot \vec{R} + \varphi_i \right) \tag{14}$$

where $a_i$, $\Omega_i$, $\varphi_i$ and $\vec{K}_i$ are the amplitude, angular frequency, phase angle and wave vector of the $i$-th regular wave, respectively. The angular frequency is $\Omega_i \in (\omega_{i-1}, \omega_i)$. The amplitude can be determined by:

$$\frac{1}{2} a_i^2(\Omega_i) = \int_{\omega_{i-1}}^{\omega_i} S(\omega) d\omega \tag{15}$$

The linear dispersion relation is considered [33]:

$$\Omega_i^2 = g\widetilde{k}_i \tanh \widetilde{k}_i H_{bed} \tag{16}$$

where $g$ is gravity. The wave number $\widetilde{k}_i = \left| \vec{K}_i \right|$. Based on Equation (16), the wave number is obtained. Further, the wave length $\lambda_i$ can be determined via the relation between the wave number $\widetilde{k}_i$ and the wave length $\widetilde{k}_i = 2\pi/\lambda_i$. Based on Equations (14)–(16) and Figure 4, and letting $n = 6$, the irregular wave is simulated by regular waves and listed in Table 1. It is assumed that the total wave energy flow rate of the regular waves is equal to that of the Jonswap wave spectrum. In cases 1~4, the numbers of regular waves range from 3 to 6. Every energy flow rate of every regular wave is assumed to be the same. According to Equation (15), the amplitude of each regular wave is the same, and there is no distinction

between the dominated wave and the secondary wave. Moreover, the frequency of the regular wave and the given peak frequency are significantly different. In case 5, when six regular waves are used to simulate an irregular wave, every energy flow rate of every regular wave is different. It is obtained that the dominated wave frequency is consistent with the given peak frequency, and the amplitude of the dominated wave is significantly larger than that of other waves. The simulated results of case 6 are used later.

**Table 1.** Irregular wave simulated by regular waves [$H_s$ = 15.4 m, $P_w$ = 16.5 s, $n$ = 6, $H_{bed}$ = 1300 m].

| Case | Number of Regular Waves | | 1 | 2 | 3 | 4 | 5 | 6 |
|---|---|---|---|---|---|---|---|---|
| 1 | 3 | $a_i$ (m) | 2.603 | 2.603 | 2.603 | - | - | - |
| | | $f_i$ (Hz) | 0.0369 | 0.0390 | 0.1893 | - | - | - |
| 2 | 4 | $a_i$ (m) | 2.255 | 2.255 | 2.255 | 2.255 | - | - |
| | | $f_i$ (Hz) | 0.0365 | 0.0382 | 0.0390 | 0.0398 | - | - |
| 3 | 5 | $a_i$ (m) | 2.017 | 2.017 | 2.017 | 2.017 | 2.017 | - |
| | | $f_i$ (Hz) | 0.0365 | 0.0382 | 0.0390 | 0.0398 | 0.0406 | - |
| 4 | 6 | $a_i$ (m) | 1.841 | 1.841 | 1.841 | 1.841 | 1.841 | 1.841 |
| | | $f_i$ (Hz) | 0.0365 | 0.0382 | 0.0390 | 0.0398 | 0.0406 | 0.1901 |
| 5 | 6 | $a_i$ (m) | 1.142 | 4.208 | 2.630 | 1.364 | 0.843 | 0.605 |
| | | $f_i$ (Hz) | 0.0425 | 0.0600 | 0.0850 | 0.1150 | 0.1500 | 0.2664 |
| | | $\tilde{k}_i$(1/m) | 0.0073 | 0.0145 | 0.0291 | 0.0533 | 0.0906 | 0.2859 |
| | | $\lambda_i$ (m) | 861.5 | 433.3 | 215.9 | 117.9 | 69.3 | 22.0 |

*2.3. Dynamic Equilibrium with the Effects of the Steady Current and Irregular Wave*

The dynamic equilibrium in the vertical direction for pontoon 3 is:

$$M_3\ddot{x}_{3d} - F_{B3} + W_3 + T_C = 0 \tag{17}$$

where $M_3$ is the mass of pontoon 3. $T_C$ is the tension of rope C. Substituting Equations (2) and (10) into Equation (17), one obtains:

$$M_3\ddot{x}_{3d} + T_{Cd} - F_{B3d} = 0 \tag{18}$$

where the dynamic tension of the rope C is:

$$T_{Cd} = K_{Cd}(x_{3d} - x_{1d}) \tag{19}$$

in which $K_{Cd}$ is the effective spring constant. $x_{3d} - x_{1d}$ is the dynamic elongation between floating platform 1 and pontoon 3. Considering the safety of the rope, some buffer springs are used to serially connect the rope between elements 1 and 3. The effective spring constant of the rope–buffer spring connection is obtained:

$$K_{Cd} = \frac{K_{\text{rope C}}}{1 + K_{\text{rope C}}/K_{C,spring}} \tag{20}$$

where $K_{C,spring}$ is the constant of the spring connecting with rope C. The effective spring constant of the rope C, $K_{\text{rope C}} = E_C A_C / L_C$, where $E_C$, $A_C$, and $L_C$ are the Young's modulus, cross-sectional area and length of rope C, respectively.

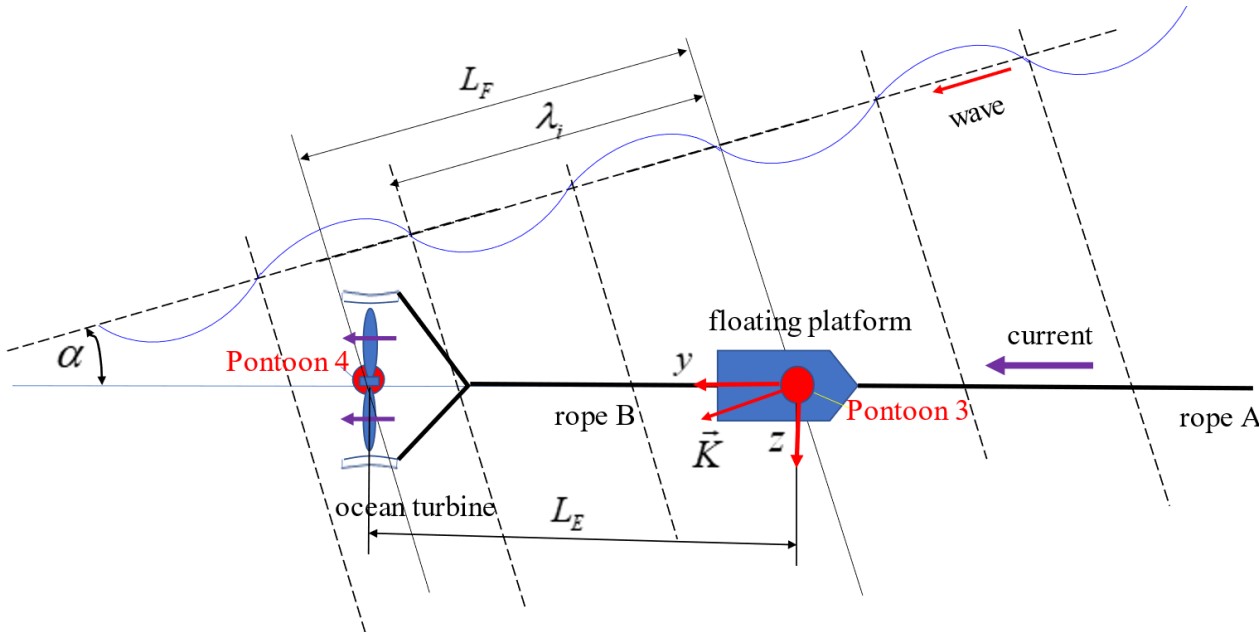

**Figure 4.** Relation among phase $f_i$, wave length $\lambda_i$ and relative direction $\alpha$ of wave and current and distance $L_E$ between the two pontoons.

Assuming the coordinates at pontoon 3 are as shown in Figure 3:

$$\overrightarrow{R}_{\text{pontoon 3}} = 0, \tag{21}$$

The sea surface elevation at pontoon 3 is:

$$x_{w,\text{pontoon 3}} = \sum_{i=1}^{N=6} a_i \sin(\Omega_i t + \varphi_i) \tag{22}$$

The coordinates at pontoon 4 are as shown in Figure 4:

$$\overrightarrow{R}_{\text{pontoon 4}} = L_E \overrightarrow{j} \tag{23}$$

The sea surface elevation at pontoon 4 is:

$$x_{w,\text{pontoon 4}} = \sum_{i=1}^{N=6} a_i \sin(\Omega_i t + \varphi_i + \phi_i) \tag{24}$$

where the phase angle $\phi_i = \frac{2\pi L_E}{\lambda_i} \cos\alpha$ and $L_E = \sqrt{L_B^2 - (L_C - L_D)^2}$. The values of the relative angle $\alpha$ and the wave length $\lambda_i$ are naturally determined. Nevertheless, the length $L_E$ can be changed to obtain the desired phase angle $\phi_i$.

The wave force on the pontoons should include horizontal force and vertical force. Because the length of the ropes connecting the pontoon to the turbine and the carrier is more than 60 m, and the rope can only transmit the axial force and cannot transmit the lateral force, the effect of horizontal force on the dynamic stability of system can be ignored. The volume of the surfaced pontoon should be reduced as much as possible to reduce the wave force, which can increase the dynamic stability and safety of the system and can also be analyzed by the one-way-coupled FSI method. Because the volume of the pontoon is considered small, the horizontal wave force to the pontoon is small. In addition, the length of the ropes connecting the pontoon to the turbine and the carrier is more than 60 m, and the rope can only transmit the axial force and cannot transmit the lateral force; the effect

of horizontal force on the dynamic stability of system can be ignored. The corresponding dynamic vertical buoyance of pontoon 3 can be expressed as:

$$F_{B3d}(t) = \sum_{i=1}^{N} [f_{Bs,i} \sin \Omega_i t + f_{Bc,i} \cos \Omega_i t] - A_{Bx} \rho g x_{3d} \tag{25}$$

where $f_{Bs,i} = A_{Bx} \rho g a_i \cos \varphi_i$, $f_{Bc,i} = A_{Bx} \rho g a_i \sin \varphi_i$. Substituting Equations (19) and (22) into Equation (18), one obtains:

$$M_3 \ddot{x}_{3d} - K_{Cd} x_{1d} + (K_{Cd} + A_{Bx} \rho g) x_{3d} = \sum_{i=1}^{N} [f_{Bs,i} \sin \Omega_i t + f_{Bc,i} \cos \Omega_i t] \tag{26}$$

The pontoon is composed of two parts: (1) the floating section on the water surface and (2) the underwater container. The floating section on the water surface is cylindrical with equal diameters, and the dynamic buoyancy of the pontoon is related to its dynamic displacement. The mass of the pontoon can be controlled by the water in the underwater container. Therefore, the mass and cross-sectional area of the pontoon can be considered, and their independent, individual effects can be studied.

The dynamic equilibrium in the vertical direction for pontoon 4 is:

$$M_4 \ddot{x}_{4d} - F_{B4} + W_4 + T_D = 0 \tag{27}$$

where $M_4$ is the mass of the pontoon 4, and $T_D$ is the tension of the rope D. Substituting Equations (2) and (11) into Equation (29), one obtains:

$$M_4 \ddot{x}_{4d} - F_{B4d} + T_{Dd} = 0 \tag{28}$$

where the dynamic tension of rope D is:

$$T_{Dd} = K_{Dd}(x_{4d} - x_{2d}) \tag{29}$$

in which $K_{Dd}$ is the effective spring constant. $x_{4d} - x_{2d}$ is the dynamic elongation between floating platform 2 and pontoon 4. Considering the safety of the rope, some buffer springs are used to serially connect the rope between elements 2 and 4. The effective spring constant of the rope–buffer spring connection is obtained:

$$K_{Dd} = \frac{K_{\text{rope D}}}{1 + K_{\text{rope D}}/K_{D,spring}} \tag{30}$$

where $K_{D,spring}$ is the constant of the spring connecting with rope D. The effective spring constant of rope D, $K_{\text{rope D}} = E_D A_D / L_D$, where $E_D$, $A_D$, and $L_D$ are the Young's modulus, cross-sectional area and length of the rope D, respectively.

According to Equation (24), the dynamic buoyance of pontoon 4 is:

$$F_{B4d}(t) = \sum_{i=1}^{N} [f_{Ts,i} \sin \Omega_i t + f_{Tc,i} \cos \Omega_i t] - A_{BT} \rho g x_{4d} \tag{31}$$

where $f_{Ts,i} = A_{BT} \rho g a_i \cos(\varphi_i + \phi_i)$, $f_{Tc,i} = A_{BT} \rho g a_i \sin(\varphi_i + \phi_i)$. Substituting Equations (29) and (31) into Equation (28), one obtains:

$$M_4 \ddot{x}_{4d} - K_{Dd} x_{2d} + (K_{Dd} + A_{BT} \rho g) x_{4d} = \sum_{i=1}^{N} [f_{Ts,i} \sin \Omega_i t + f_{Tc,i} \cos \Omega_i t] \tag{32}$$

The dynamic equilibrium in the vertical direction for the floating platform is:

$$\left( M_1 + m_{eff,x} \right) \ddot{x}_{1d} - F_{B1s} + W_1 - T_C + T_A \sin \theta_A + T_B \sin \theta_B = 0 \tag{33}$$

where $M_1$ is the mass of the platform. The dynamic, effective mass of rope 1 in the x-direction, $m_{eff,x} = \frac{4f_g L_{As} \sin \theta_1}{\pi^2}$, was derived by Lin and Chen [15]. Substituting Equations (2) and (7) into Equation (33), one obtains:

$$\left( M_1 + m_{eff,x} \right) \ddot{x}_{1d} + T_{Cd} + T_{Ad} \sin \theta_A + T_{Bd} \sin \theta_B = 0 \tag{34}$$

where $T_A$ is the dynamic tension of rope A.

$$T_{Ad} = K_{Ad} \delta_{Ad} \tag{35}$$

The dynamic elongation is $\delta_{Ad} = L_{Ad} - L_A$ where $L_A$ and $L_{Ad}$ are the static and dynamic lengths of rope A, respectively. The effective spring constant of the rope–buffer spring connection is:

$$K_{Ad} = \frac{K_{\text{rope A}}}{1 + K_{\text{rope A}} / K_{A,spring}} \tag{36}$$

where $K_{A,spring}$ is the constant of the spring connecting to rope A. The effective spring constant of rope A is $K_{\text{rope A}} = E_A A_A / L_A$, where $E_A$ and $A_A$ are the Young's modulus and the cross-sectional area of rope A, respectively. The static and dynamic lengths are:

$$L_A = \sqrt{x_{1s}^2 + y_{1s}^2}, \; L_{Ad} = \sqrt{(x_{1s} + x_{1d})^2 + (y_{1s} + y_{1d})^2} \tag{37}$$

The approximated dynamic elongation is proposed by using the Taylor formula:

$$\delta_{Ad} = \frac{x_{1s}}{L_A} x_{1d} + \frac{y_{1s}}{L_A} y_{1d} \tag{38}$$

where the dynamic tension of rope B is:

$$T_{Bd} = K_{Bd} \delta_{Bd} \tag{39}$$

and where the dynamic elongation $\delta_{Bd} = L_{Bd} - L_{Bs}$. $L_{Bs}$ and $L_{Bd}$ are the static and dynamic lengths of rope B. The effective spring constant of the rope–buffer spring connection is:

$$K_{Bd} = \frac{K_{\text{rope B}}}{1 + K_{\text{rope B}} / K_{B,spring}} \tag{40}$$

where $K_{B,spring}$ is the constant of the spring connecting with rope B. The effective spring constant of rope B is $K_{\text{rope B}} = E_B A_B / L_B$, in which $E_B$ and $A_B$ are the Young's modulus and the cross-sectional area of rope B. The static and dynamic lengths are:

$$L_B = \sqrt{(x_{1s} - x_{2s})^2 + (y_{1s} - y_{2s})^2}, \; L_{Bd} = \sqrt{(x_1 - x_2)^2 + (y_1 - y_2)^2} \tag{41}$$

Using the Tylor formula, one can obtain the approximated dynamic elongation:

$$\delta_{Bd} = \frac{x_{1s} - x_{2s}}{L_B} (x_{1d} - x_{2d}) + \frac{y_{1s} - y_{2s}}{L_B} (y_{1d} - y_{2d}) \tag{42}$$

Substituting Equations (19), (35), (38), (39) and (42) into Equation (34), one obtains:

$$\begin{aligned} \left( M_1 + m_{eff,x} \right) \ddot{x}_{1d} + \left( K_{Ad} \frac{x_{1s}}{L_A} \sin \theta_A + K_{Bd} \frac{x_{1s} - x_{2s}}{L_B} \sin \theta_B - K_{Cd} \right) x_{1d} - \left( K_{Bd} \frac{x_{1s} - x_{2s}}{L_B} \sin \theta_B \right) x_{2d} \\ + K_{Cd} x_{3d} + \left( K_{Ad} \frac{y_{1s}}{L_A} \sin \theta_A + K_{Bd} \frac{y_{1s} - y_{2s}}{L_B} \sin \theta_B \right) y_{1d} - \left( K_{Bd} \frac{y_{1s} - y_{2s}}{L_B} \sin \theta_B \right) y_{2d} = 0 \end{aligned} \tag{43}$$

The dynamic equilibrium in the vertical direction for the turbine is:

$$- M_2 \ddot{x}_{2d} - W_2 + F_{B2s} + T_D + T_B \sin \theta_B = 0 \tag{44}$$

Substituting Equations (2) and (9) into Equation (44), one obtains:

$$-M_2\ddot{x}_{2d} + T_{Dd} + T_{Bd}\sin\theta_B = 0 \tag{45}$$

Substituting Equations (29), (39) and (42) into Equation (45), one obtains:

$$
\begin{aligned}
M_2\ddot{x}_{2d} - K_{Bd}\tfrac{x_{1s}-x_{2s}}{L_B}\sin\theta_B x_{1d} + \left(K_{Dd} + K_{Bd}\tfrac{x_{1s}-x_{2s}}{L_B}\sin\theta_B\right)x_{2d} - K_{Dd}x_{4d} \\
- K_{Bd}\tfrac{y_{1s}-y_{2s}}{L_B}\sin\theta_B y_{1d} + K_{Bd}\tfrac{y_{1s}-y_{2s}}{L_B}\sin\theta_B y_{2d} = 0
\end{aligned}
\tag{46}
$$

The dynamic equilibrium in the horizontal direction for the floating platform is:

$$-\left(M_1 + m_{eff,y}\right)\ddot{y}_{1d} + F_{DFy} - T_A\cos\theta_A + T_B\cos\theta_B = 0 \tag{47}$$

where $y_{1d}$ is the dynamic horizontal displacement of the floating platform. The dynamic effective mass of rope A in the y-direction is $m_{eff,y} = \frac{4 f_g L_A \cos\theta_A}{\pi^2}$ [26]. The horizontal force on the platform due to the current velocity $V$ and the horizontal velocity $\dot{y}_{1d}$ of the platform are expressed as [34]:

$$F_{DFY} = \tfrac{1}{2}C_{DFy}\rho A_{FY}\left(V - \dot{y}_{1d}\right)^2 = \tfrac{1}{2}C_{DFy}\rho A_{FY}\left(V^2 - 2V\dot{y}_{1d} + \dot{y}_{1d}^2\right) \approx F_{DFs} - C_{DFy}\rho A_{FY}V\dot{y}_{1d} \tag{48}$$

Because $\dot{y}_{1d} \ll V$, the term $\dot{y}_{1d}^2$ is negligible. The drag coefficient of the floating platform is considered close to that of a bullet, i.e., $C_{DFy} \approx 0.3$, [26].

Substituting Equations (2), (7), (35), (38), (39), (42) and (48) into Equation (47), one obtains:

$$
\begin{aligned}
\left(M_1 + m_{eff,y}\right)\ddot{y}_{1d} + C_{DFy}\rho A_{Fy}V\dot{y}_{1d} \\
+ \left(K_{Ad}\tfrac{x_{1s}}{L_A}\cos\theta_A - K_{Bd}\tfrac{x_{1s}-x_{2s}}{L_B}\cos\theta_B\right)x_{1d} + K_{Bd}\tfrac{x_{1s}-x_{2s}}{L_B}\cos\theta_B x_{2d} \\
+ \left(K_{Ad}\tfrac{y_{1s}}{L_A}\cos\theta_A - K_{Bd}\tfrac{y_{1s}-y_{2s}}{L_B}\cos\theta_B\right)y_{1d} + K_{Bd}\tfrac{y_{1s}-y_{2s}}{L_B}\cos\theta_B y_{2d} = 0
\end{aligned}
\tag{49}
$$

It is discovered from Equation (49) that the second term is the damping effect for vibration of the system. The damping effect depends on the parameters: (1) the damping coefficient $C_{DyF}$, (2) the damping area $A_{BY}$ and (3) the current velocity $V$.

The dynamic equilibrium in the horizontal direction for the turbine is:

$$-M_2\ddot{y}_{2d} + F_{DTy} - T_B\cos\theta_B = 0 \tag{50}$$

where $y_{2d}$ is the dynamic, horizontal displacement of the turbine. The horizontal force on the platform caused by the current velocity $V$ and the horizontal velocity $\dot{y}_{2d}$ of the turbine is expressed as [34]:

$$F_{DTy} = C_{DTy}\tfrac{1}{2}\rho A_{Ty}\left(V - \dot{y}_{2d}\right)^2 = C_{DTy}\tfrac{1}{2}\rho A_{Ty}\left(V^2 - 2V\dot{y}_{2d} + \dot{y}_{2d}^2\right) \approx F_{DTs} - C_{DyT}\rho A_{Ty}V\dot{y}_{2d} \tag{51}$$

where $A_{Ty}$ is the effective operating area of the turbine. The theoretical effective drag coefficient of optimum efficiency is $C_{DTy} = 8/9$, [27]. Considering $\dot{y}_{2d} \ll V$, the term $\dot{y}_{2d}^2$ is negligible. Substituting Equations (2), (8), (39), (42) and (51) into Equation (50), one obtains:

$$
\begin{aligned}
M_2\ddot{y}_{2d} + C_{DTy}\rho A_{Ty}V\dot{y}_{2d} + K_{Bd}\tfrac{x_{1s}-x_{2s}}{L_B}\cos\theta_B x_{1d} \\
- K_{Bd}\tfrac{x_{1s}-x_{2s}}{L_B}\cos\theta_B x_{2d} + K_{Bd}\tfrac{y_{1s}-y_{2s}}{L_B}\cos\theta_B y_{1d} - K_{Bd}\tfrac{y_{1s}-y_{2s}}{L_B}\cos\theta_B y_{2d} = 0
\end{aligned}
\tag{52}
$$

It is discovered from Equation (52) that the second term is the damping effect for vibration of the system. The damping effect depends on the parameters: (1) the damping coefficient $C_{DTy}$, (2) the damping area $A_{TY}$ and (3) the current velocity $V$.

Finally, the coupled equations of motion in terms of the dynamic displacements $x_{1d}$, $x_{2d}$, $x_{3d}$, $x_{4d}$, $y_{1d}$, and $y_{2d}$ are discovered as Equations (26), (32), (34), (49), (52) and (56).

### 2.4. Solution Method

2.4.1. Free Vibration

Without the excitation of wave and under steady ocean current, the coupled motion of the system is in free vibration. According to Equations (26), (32), (34), (49), (52) and (56), the coupled equations of free vibration can be expressed as:

$$\mathbf{M}\ddot{\mathbf{Z}}_d + \mathbf{C}\dot{\mathbf{Z}}_d + \mathbf{K}\mathbf{Z}_d = 0 \tag{53}$$

where

$$\mathbf{Z}_d = \begin{bmatrix} x_{1d} \\ x_{2d} \\ x_{3d} \\ x_{4d} \\ y_{1d} \\ y_{2d} \end{bmatrix}, \ \mathbf{M} = \begin{bmatrix} \left(M_1 + m_{eff,x}\right) & 0 & 0 & 0 & 0 & 0 \\ 0 & M_2 & 0 & 0 & 0 & 0 \\ 0 & 0 & M_3 & 0 & 0 & 0 \\ 0 & 0 & 0 & M_4 & 0 & 0 \\ 0 & 0 & 0 & 0 & \left(M_1 + m_{eff,y}\right) & 0 \\ 0 & 0 & 0 & 0 & 0 & M_2 \end{bmatrix} \tag{54}$$

$$\mathbf{C} = \begin{bmatrix} C_1 & 0 & 0 & 0 & 0 & 0 \\ 0 & C_2 & 0 & 0 & 0 & 0 \\ 0 & 0 & C_3 & 0 & 0 & 0 \\ 0 & 0 & 0 & C_4 & 0 & 0 \\ 0 & 0 & 0 & 0 & C_5 & 0 \\ 0 & 0 & 0 & 0 & 0 & C_6 \end{bmatrix}, \ \mathbf{K} = \begin{bmatrix} K_{11} & K_{12} & K_{13} & 0 & K_{15} & K_{16} \\ K_{21} & K_{22} & 0 & K_{24} & K_{25} & K_{26} \\ K_{31} & 0 & K_{33} & 0 & 0 & 0 \\ 0 & K_{42} & 0 & K_{44} & 0 & 0 \\ K_{51} & K_{52} & 0 & 0 & K_{55} & K_{56} \\ K_{61} & K_{62} & 0 & 0 & K_{65} & K_{66} \end{bmatrix}$$

$$C_1 = C_2 = C_3 = C_4 = 0, \ C_5 = C_{DFy}\rho A_{Fy}V, \ C_6 = C_{DTy}\rho A_{Ty}V$$

$$K_{11} = \left(K_{Ad}\frac{x_{1s}}{L_A}\sin\theta_A + K_{Bd}\frac{x_{1s}-x_{2s}}{L_B}\sin\theta_B + K_{Cd}\right)$$

$$K_{12} = -\left(K_{Bd}\frac{x_{1s}-x_{2s}}{L_B}\sin\theta_B\right), \ K_{13} = -K_{Cd}$$

$$K_{15} = \left(K_{Ad}\frac{y_{1s}}{L_A}\sin\theta_A + K_{Bd}\frac{y_{1s}-y_{2s}}{L_B}\sin\theta_B\right), \ K_{16} = -\left(K_{Bd}\frac{y_{1s}-y_{2s}}{L_B}\sin\theta_B\right)$$

$$K_{21} = -K_{Bd}\frac{x_{1s}-x_{2s}}{L_B}\sin\theta_B, \ K_{22} = \left(K_{Dd} + K_{Bd}\frac{x_{1s}-x_{2s}}{L_B}\sin\theta_B\right),$$

$$K_{24} = -K_{Dd}, \ K_{25} = -K_{Bd}\frac{y_{1s}-y_{2s}}{L_B}\sin\theta_B, \ K_{26} = K_{Bd}\frac{y_{1s}-y_{2s}}{L_B}\sin\theta_B \tag{55}$$

$$K_{31} = -K_{Cd} \qquad\qquad K_{42} = -K_{Dd}$$

$$K_{33} = (K_{Cd} + A_{Bx}\rho g) \qquad K_{44} = (K_{Dd} + A_{BT}\rho g)$$

$$K_{51} = \left(K_{Ad}\frac{x_{1s}}{L_A}\cos\theta_A - K_{Bd}\frac{x_{1s}-x_{2s}}{L_B}\cos\theta_B\right), \ K_{52} = K_{Bd}\frac{x_{1s}-x_{2s}}{L_B}\cos\theta_B$$

$$K_{55} = \left(K_{Ad}\frac{y_{1s}}{L_A}\cos\theta_A - K_{Bd}\frac{y_{1s}-y_{2s}}{L_B}\cos\theta_B\right), \ K_{56} = K_{Bd}\frac{y_{1s}-y_{2s}}{L_B}\cos\theta_B$$

$$K_{61} = K_{Bd}\frac{x_{1s}-x_{2s}}{L_B}\cos\theta_B, \ K_{62} = -K_{Bd}\frac{x_{1s}-x_{2s}}{L_B}\cos\theta_B,$$

$$K_{65} = K_{Bd}\frac{y_{1s}-y_{2s}}{L_B}\cos\theta_B, \ K_{66} = -K_{Bd}\frac{y_{1s}-y_{2s}}{L_B}\cos\theta_B;$$

The solution of Equation (53) is assumed to be:

$$\mathbf{Z}_d = \begin{bmatrix} \bar{x}_{1d} & \bar{x}_{2d} & \bar{x}_{3d} & \bar{x}_{4d} & \bar{y}_{1d} & \bar{y}_{2d} \end{bmatrix}^{\mathrm{T}} = (\bar{\mathbf{z}}_{\mathbf{dc}}\cos\Omega t + \bar{\mathbf{z}}_{\mathbf{ds}}\sin\Omega t) \tag{56}$$

where $\bar{\mathbf{z}}_{\mathbf{dc}} = \begin{bmatrix} \bar{x}_{1d,c} & \bar{x}_{2d,c} & \bar{x}_{3d,c} & \bar{x}_{4d,c} & \bar{y}_{1d,c} & \bar{y}_{2d,c} \end{bmatrix}^{\mathrm{T}}$,
$\bar{\mathbf{z}}_{\mathbf{ds}} = \begin{bmatrix} \bar{x}_{1d,s} & \bar{x}_{2d,s} & \bar{x}_{3d,s} & \bar{x}_{4d,s} & \bar{y}_{1d,s} & \bar{y}_{2d,s} \end{bmatrix}^{\mathrm{T}}$. Substituting Equation (56) into Equation (53), one obtains:

$$\left(\left(\mathbf{M}^{-1}\mathbf{K} - \Omega^2\mathbf{I}\right)\bar{\mathbf{z}}_{\mathbf{dc}} + \Omega\mathbf{M}^{-1}\mathbf{C}\bar{\mathbf{z}}_{\mathbf{ds}}\right)\cos\Omega t$$
$$+ \left(\left(\mathbf{M}^{-1}\mathbf{K} - \Omega^2\mathbf{I}\right)\bar{\mathbf{z}}_{\mathbf{ds}} - \Omega\mathbf{M}^{-1}\mathbf{C}\bar{\mathbf{z}}_{\mathbf{dc}}\right)\sin\Omega t = 0 \tag{57}$$

Due to the orthogonality of $\sin\Omega t$ and $\cos\Omega t$, Equation (57) becomes:

$$\left(\mathbf{M}^{-1}\mathbf{K} - \Omega^2\mathbf{I}\right)\bar{\mathbf{z}}_{\mathbf{dc}} + \Omega\mathbf{M}^{-1}\mathbf{C}\bar{\mathbf{z}}_{\mathbf{ds}} = 0 \tag{58}$$

$$- \Omega \mathbf{M}^{-1} \mathbf{C} \bar{\mathbf{z}}_{\mathbf{d}c} + \left( \mathbf{M}^{-1} \mathbf{K} - \Omega^2 \mathbf{I} \right) \bar{\mathbf{z}}_{\mathbf{d}s} = 0 \tag{59}$$

Further, Equation (58) can be expressed as:

$$\bar{\mathbf{z}}_{\mathbf{d}c} = - \left( \mathbf{M}^{-1} \mathbf{K} - \Omega^2 \mathbf{I} \right)^{-1} \Omega \mathbf{M}^{-1} \mathbf{C} \bar{\mathbf{z}}_{\mathbf{d}s} \tag{60}$$

Substituting Equation (60) into Equation (59), one obtains:

$$\mathbf{Q} \bar{\mathbf{z}}_{\mathbf{d}s} = 0 \tag{61}$$

where $\mathbf{Q} = \Omega^2 \mathbf{M}^{-1} \mathbf{C} \left( \mathbf{M}^{-1} \mathbf{K} - \Omega^2 \mathbf{I} \right)^{-1} \mathbf{M}^{-1} \mathbf{C} + \left( \mathbf{M}^{-1} \mathbf{K} - \Omega^2 \mathbf{I} \right)$. The frequency equation is:

$$|\mathbf{Q}| = 0 \tag{62}$$

The natural frequencies of the system can be determined via Equation (62).

### 2.4.2. Forced Vibration

Considering the excitation of the wave, the coupled Equations (26), (32), (34), (49), (52) and (56) can be rewritten in the matrix format as follows:

$$\mathbf{M} \ddot{\mathbf{Z}}_d + \mathbf{C} \dot{\mathbf{Z}}_d + \mathbf{K} \mathbf{Z}_d = \mathbf{F}_d \tag{63}$$

where

$$\mathbf{F}_d = \left[ \begin{array}{cccccc} 0 & 0 & \sum\limits_{i=1}^{N} [f_{Bs,i} \sin \Omega_i t + f_{Bc,i} \cos \Omega_i t] & \sum\limits_{i=1}^{N} [f_{Ts,i} \sin \Omega_i t + f_{Tc,i} \cos \Omega_i t] & 0 & 0 \end{array} \right]^T,$$

$$f_{Bs,i} = A_{Bx} \rho g a_i \cos \varphi_i, \ f_{Bc,i} = A_{Bx} \rho g a_i \sin \varphi_i$$

$$f_{Ts,i} = A_{BT} \rho g a_i \cos(\varphi_i + \phi_i), \ f_{Tc,i} = A_{BT} \rho g a_i \sin(\varphi_i + \phi_i) \tag{64}$$

The solution of Equation (63) is assumed:

$$\mathbf{Z}_d = \left[ \begin{array}{cccccc} x_{1d} & x_{2d} & x_{3d} & x_{4d} & y_{1d} & y_{2d} \end{array} \right]^{\mathrm{T}} = \sum_{i=1}^{N} (\mathbf{z}_{\mathbf{d}c,i} \cos \Omega_i t + \mathbf{z}_{\mathbf{d}s,i} \sin \Omega_i t), \tag{65}$$

where $\mathbf{z}_{\mathbf{d}c,i} = \left[ \begin{array}{cccccc} x_{1d,c} & x_{2d,c} & x_{3d,c} & x_{4d,c} & y_{1d,c} & y_{2d,c} \end{array} \right]^{\mathrm{T}}$, $\mathbf{z}_{\mathbf{d}s,i} = \left[ \begin{array}{cccccc} x_{1d,s} & x_{2d,s} & x_{3d,s} & x_{4d,s} & y_{1d,s} & y_{2d,s} \end{array} \right]^{\mathrm{T}}$. Substituting Equation (65) into Equation (63), one obtains:

$$- \sum_{i=1}^{N} \Omega_i^2 (\mathbf{z}_{\mathbf{d}c,i} \cos \Omega_i t + \mathbf{z}_{\mathbf{d}s,i} \sin \Omega_i t) + \mathbf{M}^{-1} \mathbf{C} \sum_{i=1}^{N} (-\Omega_i \mathbf{z}_{\mathbf{d}c,i} \sin \Omega_i t + \Omega_i \mathbf{z}_{\mathbf{d}s,i} \cos \Omega_i t)$$

$$+ \mathbf{M}^{-1} \mathbf{K} \sum_{i=1}^{N} (\mathbf{z}_{\mathbf{d}c,i} \cos \Omega_i t + \mathbf{z}_{\mathbf{d}s,i} \sin \Omega_i t) = \sum_{i=1}^{N} (\mathbf{F}_{s,i} \sin \Omega_i t + \mathbf{F}_{c,i} \cos \Omega_i t) \tag{66}$$

Multiplying Equation (66) by $\cos \Omega_m t$ and integrating it from 0 to the period $T_m$, $2\pi/\Omega_m$, Equation (66) becomes:

$$\sum_{i=1}^{N} \mathbf{a}_{im} \mathbf{z}_{\mathbf{d}c,i} + \sum_{i=1}^{N} \mathbf{b}_{im} \mathbf{z}_{\mathbf{d}s,i} = \chi_{cm}, \ m = 1, 2, \dots, N \tag{67}$$

where

$$\mathbf{a}_{im} = \left[\alpha_{im}\left(\mathbf{M}^{-1}\mathbf{K} - \Omega_i^2\mathbf{I}\right) - \beta_{im}\Omega_i\mathbf{M}^{-1}\mathbf{C}\right], \mathbf{b}_{im} = \left[\beta_{im}\left(\mathbf{M}^{-1}\mathbf{K} - \Omega_i^2\mathbf{I}\right) - \alpha_{im}\Omega_i\mathbf{M}^{-1}\mathbf{C}\right]$$

$$\chi_{cm} = \sum_{i=1}^{N}\left(\mathbf{F}_{s,i}\beta_{im} + \mathbf{F}_{c,i}\alpha_{im}\right)$$

$$\alpha_{im} = \begin{cases} \frac{T_m}{2}, & i = m \\ \frac{\Omega_i\sin(\Omega_i T_m)}{(\Omega_i+\Omega_m)(\Omega_i-\Omega_m)}, & i \neq m \end{cases}, \quad \beta_{im} = \begin{cases} 0, & i = m \\ \frac{\Omega_i(1-\cos(\Omega_i T_m))}{(\Omega_i+\Omega_m)(\Omega_i-\Omega_m)}, & i \neq m \end{cases} \tag{68}$$

Multiplying Equation (66) by $\sin\Omega_m t$ and integrating it from 0 to the period $T_m$, $2\pi/\Omega_m$, Equation (66) becomes:

$$\sum_{i=1}^{N}\mathbf{c}_{im}\mathbf{z}_{\mathbf{d}c,i} + \sum_{i=1}^{N}\mathbf{d}_{im}\mathbf{z}_{\mathbf{d}s,i} = \chi_{sm}, \; m = 1, 2, \ldots, N \tag{69}$$

where

$$\mathbf{c}_{im} = \left[\beta_{mi}\left(\mathbf{M}^{-1}\mathbf{K} - \Omega_i^2\mathbf{I}\right) - \gamma_{im}\Omega_i\mathbf{M}^{-1}\mathbf{C}\right], \mathbf{d}_{im} = \left[\gamma_{im}\left(\mathbf{M}^{-1}\mathbf{K} - \Omega_i^2\mathbf{I}\right) - \beta_{mi}\Omega_i\mathbf{M}^{-1}\mathbf{C}\right]$$

$$\chi_{sm} = \sum_{i=1}^{N}\left(\mathbf{F}_{s,i}\gamma_{im} + \mathbf{F}_{c,i}\beta_{mi}\right)$$

$$\gamma_{im} = \begin{cases} \frac{T_m}{2}, & i = m \\ \frac{\Omega_m\sin(\Omega_i T_m)}{(\Omega_i+\Omega_m)(\Omega_i-\Omega_m)}, & i \neq m \end{cases} \tag{70}$$

Equations (67) and (69) can be written as:

$$\mathbf{BZ=F} \tag{71}$$

where

$$\mathbf{B} = \begin{bmatrix} \begin{bmatrix} \mathbf{a}_{11} & \mathbf{a}_{21} & \cdots & \mathbf{a}_{N1} \\ \mathbf{a}_{12} & \mathbf{a}_{22} & \cdots & \mathbf{a}_{N2} \\ \vdots & \vdots & \cdots & \vdots \\ \mathbf{a}_{1N} & \mathbf{a}_{2N} & \cdots & \mathbf{a}_{NN} \end{bmatrix}_{6N\times6N} & \begin{bmatrix} \mathbf{b}_{11} & \mathbf{b}_{21} & \cdots & \mathbf{b}_{N1} \\ \mathbf{b}_{12} & \mathbf{b}_{22} & \cdots & \mathbf{b}_{N2} \\ \vdots & \vdots & \cdots & \vdots \\ \mathbf{b}_{1N} & \mathbf{b}_{2N} & \cdots & \mathbf{b}_{NN} \end{bmatrix}_{6N\times6N} \\ \begin{bmatrix} \mathbf{c}_{11} & \mathbf{c}_{21} & \cdots & \mathbf{c}_{N1} \\ \mathbf{c}_{12} & \mathbf{c}_{22} & \cdots & \mathbf{c}_{N2} \\ \vdots & \vdots & \cdots & \vdots \\ \mathbf{c}_{1N} & \mathbf{c}_{2N} & \cdots & \mathbf{c}_{NN} \end{bmatrix}_{6N\times6N} & \begin{bmatrix} \mathbf{d}_{11} & \mathbf{d}_{21} & \cdots & \mathbf{d}_{N2} \\ \mathbf{d}_{12} & \mathbf{d}_{22} & \cdots & \mathbf{d}_{N2} \\ \vdots & \vdots & \cdots & \vdots \\ \mathbf{d}_{1N} & \mathbf{d}_{2N} & \cdots & \mathbf{d}_{NN} \end{bmatrix}_{6N\times6N} \end{bmatrix}_{12N\times12N}, \tag{72}$$

$$\mathbf{Z} = \begin{bmatrix} \begin{bmatrix} \mathbf{z}_{\mathbf{d}c,1} \\ \mathbf{z}_{\mathbf{d}c,2} \\ \vdots \\ \mathbf{z}_{\mathbf{d}c,N} \end{bmatrix}_{6N\times1} \\ \begin{bmatrix} \mathbf{z}_{\mathbf{d}s,1} \\ \mathbf{z}_{\mathbf{d}s,2} \\ \vdots \\ \mathbf{z}_{\mathbf{d}s,N} \end{bmatrix}_{6N\times1} \end{bmatrix}_{12N\times1}, \quad \mathbf{F} = \begin{bmatrix} \begin{bmatrix} \chi_{c1} \\ \chi_{c2} \\ \vdots \\ \chi_{cN} \end{bmatrix}_{6N\times1} \\ \begin{bmatrix} \chi_{s1} \\ \chi_{s2} \\ \vdots \\ \chi_{sN} \end{bmatrix}_{6N\times1} \end{bmatrix}_{12N\times1}$$

The solution of Equation (65) is:

$$\mathbf{Z=B^{-1}F} \tag{73}$$

Further, one can derive the dynamic tensions of ropes under irregular wave as follows: The dynamic tension of rope A is:

$$T_{Ad} = \sum_{i=1}^{N} T_{Adc,i}\cos\Omega_i t + T_{Ads,i}\sin\Omega_i t \tag{74}$$

where $T_{Adc,i} = K_{Ad}\left(\frac{x_{1s}}{L_A}x_{1dc,i} + \frac{y_{1s}}{L_A}y_{1dc,i}\right)$, $T_{Ads,i} = K_{Ad}\left(\frac{x_{1s}}{L_A}x_{1ds,i} + \frac{y_{1s}}{L_A}y_{1ds,i}\right)$.

The dynamic tension of rope B is:

$$T_{Bd} = \sum_{i=1}^{N} T_{Bdc,i}\cos\Omega_i t + T_{Bds,i}\sin\Omega_i t \tag{75}$$

where

$$T_{Bdc,i} = K_{Bd}\left[\frac{x_{2s}-x_{1s}}{L_B}(x_{2dc,i}-x_{1dc,i}) + \frac{y_{2s}-y_{1s}}{L_B}(y_{2dc,i}-y_{1dc,i})\right],$$
$$T_{Bds,i} = K_{Bd}\left[\frac{x_{2s}-x_{1s}}{L_B}(x_{2ds,i}-x_{1ds,i}) + \frac{y_{2s}-y_{1s}}{L_B}(y_{2ds,i}-y_{1ds,i})\right].$$

The dynamic tension of rope C is:

$$T_{Cd} = \sum_{i=1}^{N} T_{Cdc,i}\cos\Omega_i t + T_{Cds,i}\sin\Omega_i t \tag{76}$$

where $T_{Cdc,i} = K_{Cd}(x_{3dc,i} - x_{1dc,i})$, $T_{Cds,i} = K_{Cd}(x_{3ds,i} - x_{1ds,i})$.

The dynamic tension of rope D is:

$$T_{Dd} = \sum_{i=1}^{N} T_{Ddc,i}\cos\Omega_i t + T_{Dds,i}\sin\Omega_i t \tag{77}$$

where $T_{Ddc,i} = K_{Dd}(x_{4dc,i} - x_{2dc,i})$, $T_{Dds,i} = K_{Dd}(x_{4ds,i} - x_{2ds,i})$.

## 3. Numerical Results and Discussion

This study investigates the dynamic response of two kinds of mooring system under the typhoon irregular wave: (1) the diving depth of the turbine $L_D$ = 60 m, the diving depth of the floating platform $L_C \geq 60$ m and (2) the diving depth of the floating platform $L_C$ = 60 m, the diving depth of the turbine $L_D \geq 60$ m. Meanwhile, the effects of several parameters on the dynamic response are investigated.

Firstly, the first kind of mooring system is investigated. Consider the conditions in Figure 5a,b: (1) the depth of seabed $H_{bed}$ = 1300 m, (2) the cross-sectional area of pontoon 3 connecting to floating platform $A_{BX}$ = 2.12 m$^2$, (3) the cross-sectional area of pontoon 4 connecting to turbine $A_{BT}$ = 2.12 m$^2$, (4) no buffer spring, (5) the ropes A, B, C and D are made of some commercial, high-strength PE dyneema; Young's modulus $E_{PE}$ = 100 GPa, weight per unit length $f_{g,PE}$ = 16.22 kg/m, diameter $D_{PE}$ = 154 mm, cross-sectional area $A_{PE}$ = 0.0186 m$^2$, fracture strength $T_{fracture}$ = 759 tons, (6) the static diving depth of the turbine $L_D$ = 60 m, (7) the horizontal distance between the turbine and floating platform $L_E$ = 100 m, (8) the inclined angle of the rope A, $\theta_A = 30°$, (9) the current velocity $V$ = 1 m/s, (10) the irregular wave is simulated by six regular waves which are listed in Table 1, (11) the wave phase angles $\varphi_i$, $i = 1, 2, \ldots, 6$ are assumed as $\{30°, 60°, 90°, 120°, 170°, 270°\}$, (12) the masses of turbine, floating platform and pontoons $M_1$ = 300 tons, $M_2$ = 838 tons, $M_3 = M_4 = 250$ tons, (13) the cross-sectional area of the floating platform and turbine $A_{FY}$ = 23 m$^2$ and $A_{TY}$ = 500 m$^2$, (14) the effective damping coefficients $C_{DFy}$ = 0.3 and $C_{DTy} = 8/9$, (15) the static axial force to turbine $F_{DTs}$ = 180 tons and (16) the relative orientation between current and wave $\alpha = 60°$.

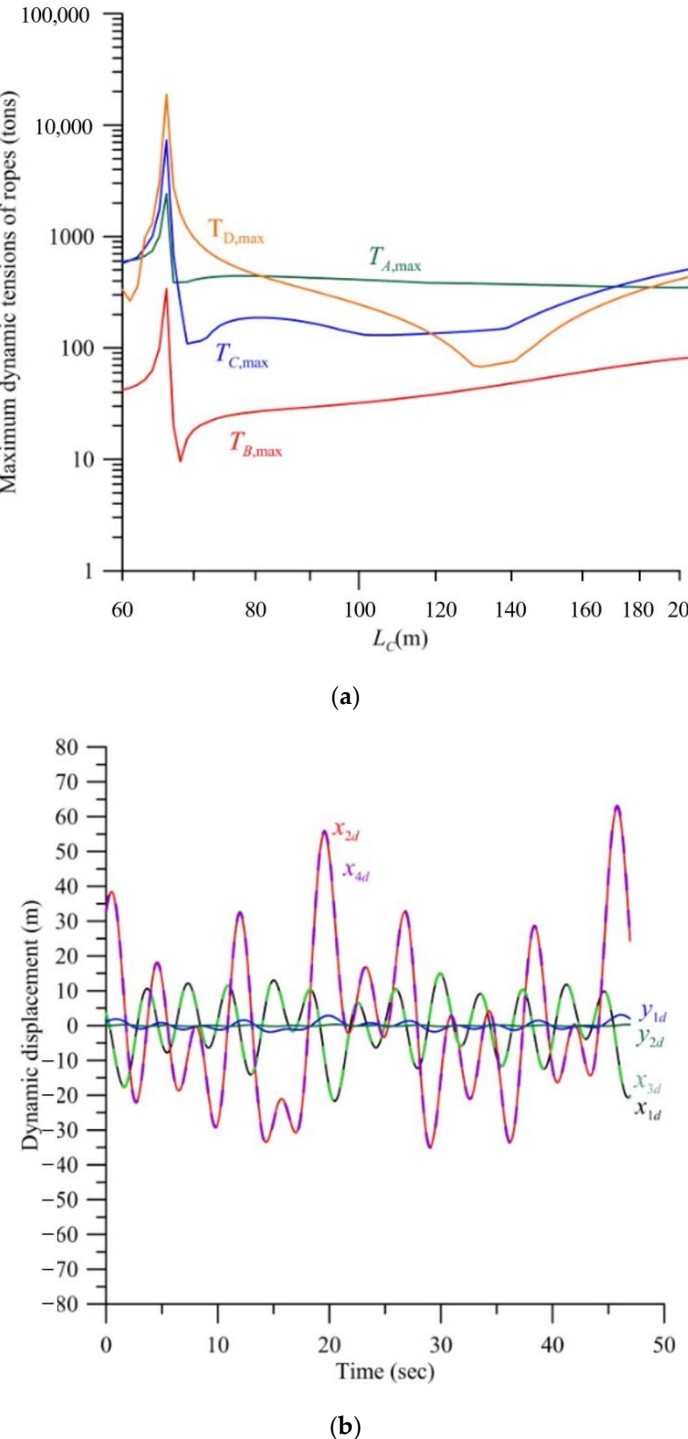

**Figure 5.** Dynamic tensions of ropes and Dynamic displacements of elements under the typhoon irregular wave. (**a**) Dynamic tension of the four ropes under the typhoon irregular wave as a function of the diving length for $L_E$ = 100 m; (**b**) Dynamic displacements of the four elements at resonance for $L_C$ = 66 m.

Figure 5a demonstrates the effect of the diving depth of the floating platform $L_C$ on the maximum dynamic tensions of the four ropes, $T_{A,max}$, $T_{B,max}$, $T_{C,max}$ and $T_{D,max}$ under the typhoon irregular wave when the diving depth of the turbine $L_D$ = 60 m. The irregular wave is simulated by six regular waves which are listed in Table 1. When the depth $L_C$ increases from 60 m, the dynamic tensions of the ropes increase significantly. If $L_{C,res}$ = 66 m , the resonance happens, and the maximum dynamic tensions $T_{A,max}$ = 2422 tons, $T_{C,max}$ = 7329 tons

and $T_{D,max}$ = 18,835 tons, which is over that of the fracture strength of rope, $T_{fracture}$ = 759 tons. Figure 5b demonstrates the vibration mode at the resonance. It is found that the displacements $x_{2d}$ and $x_{4d}$ of turbine 2 and pontoon 4 are largest. Therefore, the maximum dynamic tension is that of rope D, $T_{D,max}$.

When $L_C$ increases further, the dynamic tension decreases sharply. If $L_C$ > 80 m, all the dynamic tensions are significantly less than the fracture strength of rope, $T_{fracture}$ = 759 tons. If $L_C$ = 80 m, $T_{A,max}$ = 442 tons, $T_{B,max}$ = 27 tons, $T_{C,max}$ = 187 tons, then $T_{D,max}$ = 478 tons. The maximum one among the four dynamic tensions is $T_{max}$ = $T_{D,max}$ = 478 tons. If $L_C$ = 150 m, the maximum dynamic tension $T_{max}$ = $T_{A,max}$ = 367 tons. This is because the natural frequency changes with the length $L_C$. The excitation frequencies of the irregular wave are different to the natural frequency of the mooring system. Therefore, the resonance does not exist. It is found that the greater the diving depth of the floating platform $L_C$, the smaller the maximum dynamic tension. In other words, the mooring system of the diving depth of the floating platform $L_C$ = 150 m is better than that of $L_C$ = 80 m. Because the diving depth of the floating platform is different to that of turbine, the water flowing through the floating platform does not interfere with the flow field of the turbine. Moreover, for $L_C$ > 80 m, the dynamic tension $T_{A,max}$ of rope A decreases with the diving depth $L_C$. This is because the angle $\theta_A$ of rope A decreases with the diving depth $L_C$. The towing force is horizontal due to the ocean velocity. Meanwhile, the dynamic tension $T_{B,max}$ of rope B increases with the diving depth $L_C$. It is because the angle $\theta_B$ of rope B increases with the diving depth $L_C$.

Figure 6 presents the relation between the diving depth $L_C$ of the floating platform and the dynamic tensions of ropes under the typhoon irregular wave for the distance $L_E$ = 200 m. Aside from the distance $L_E$ =200 m, all other parameters are the same as those of Figure 5. It can be observed in Figure 6 that, when $L_E$ = 200 m, the maximum resonant position $L_{C,res}$ = 80 m is different to $L_{C,res}$ = 66 m for $L_E$ = 100 m in Figure 5. The effect of the horizontal distance between the turbine and floating platform $L_E$ on the dynamic tension with $L_C$ = 150 m is negligible.

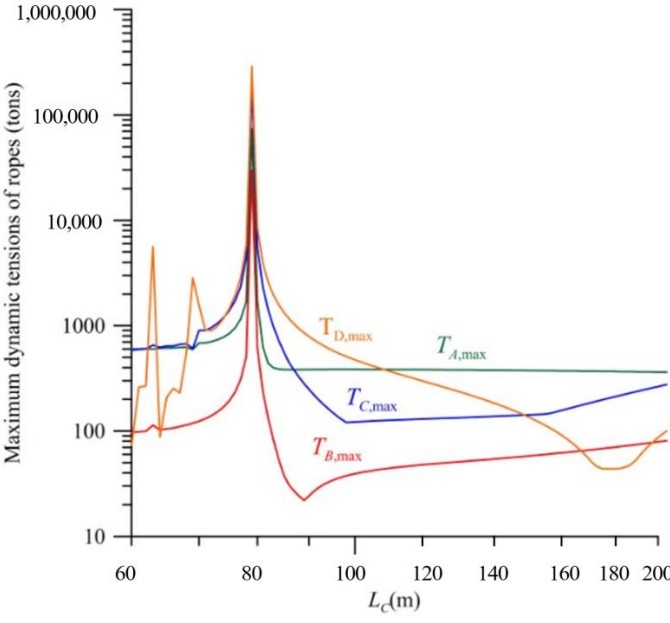

**Figure 6.** Dynamic tension of the four ropes under the typhoon irregular wave as a function of the diving length $L_C$ and the horizontal distance between the turbine and floating platform $L_E$.

Figure 7 shows the effects of the diving depth of the floating platform $L_C$ and the mass of pontoons $M_3$ and $M_4$ on the dynamic tensions of the four ropes, $T_{A,max}$, $T_{B,max}$, $T_{C,max}$ and $T_{D,max}$ under the typhoon irregular wave. In this case, the mass of pontoons $M_3$ = $M_4$ = 150 tons; other parameters are the same as those of Figure 6. It is found that, if the mass of pontoons $M_3$ = $M_4$ = 150 tons, the resonance occurs at several diving depths of the floating platform $L_C$, and the maximum dynamic tensions are over that of the fracture

strength of rope, $T_{fracture}$ = 759 tons. In other words, if the weight of the pontoon is too low, the dynamic displacement of the system is too intense, resulting in the excessive dynamic tension of the rope.

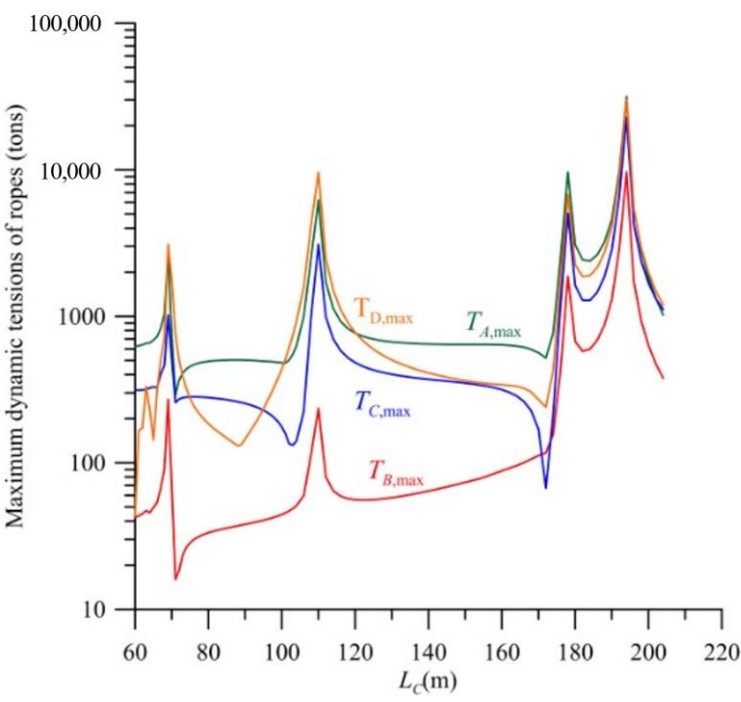

**Figure 7.** Dynamic tension of the four ropes under the typhoon irregular wave as a function of the diving length $L_C$ and the mass of pontoons $M_3$ and $M_4$.

Further, the second kind of mooring system is investigated. Figure 8 demonstrates the effect of the diving depth of the floating platform $L_D$ and the mass of pontoons $M_3$ and $M_4$ on the maximum dynamic tensions of the four ropes, $T_{A,max}$, $T_{B,max}$, $T_{C,max}$ and $T_{D,max}$ under the typhoon irregular wave when the diving depth of the turbine $L_C$ = 60 m and the horizontal distance between the turbine and floating platform $L_E$ = 100 m. All the other parameters are the same as those of Figure 5. It is found that there is no resonance. The dynamic tension increases with the diving depth of the floating platform $L_D$, especially in the case where $M_3 = M_4$ = 150 tons. The maximum tension is that of rope A, $T_{A,max}$, which is close or over that of the fracture strength of rope, $T_{fracture}$ = 759 tons. It is concluded that this mooring system should not be proposed.

Figure 9 demonstrates the effect of the diving depth of the turbine $L_D$ and the mass of pontoons $M_3$ and $M_4$ on the maximum dynamic tensions of the four ropes, $T_{A,max}$, $T_{B,max}$, $T_{C,max}$ and $T_{D,max}$ under the typhoon irregular wave when the diving depth of the floating platform $L_C$ = 60 m and the horizontal distance between the turbine and floating platform $L_E$ = 200 m. All the other parameters are the same as those in Figure 8. It is found that the maximum tension of the four ropes is the dynamic tension of rope A, $T_{A,max}$. If the mass of pontoons $M_3 = M_4$ =150 tons, the maximum tension $T_{A,max}$ decreases with the diving depth of the turbine $L_D$. However, it is the reverse for the case of the mass of pontoons $M_3 = M_4$ = 250 tons. Moreover, the dynamic tension $T_{A,max}$, with the mass of pontoons $M_3 = M_4$ = 150 tons, is obviously less than that of the mass of pontoons $M_3 = M_4$ = 250 tons.

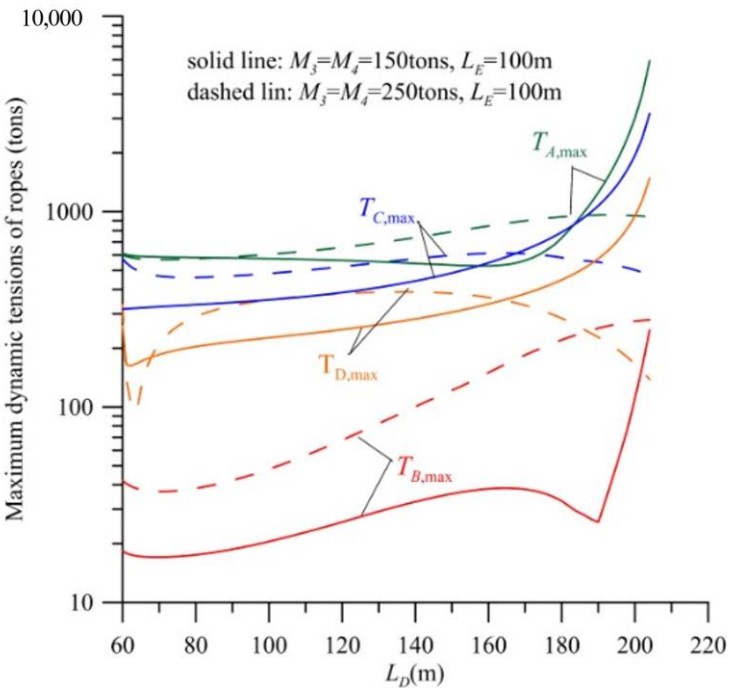

**Figure 8.** Dynamic tension of the four ropes under the typhoon irregular wave as a function of the diving length $L_D$ and the mass of pontoons $M_3$ and $M_4$ for $L_E$ = 100 m.

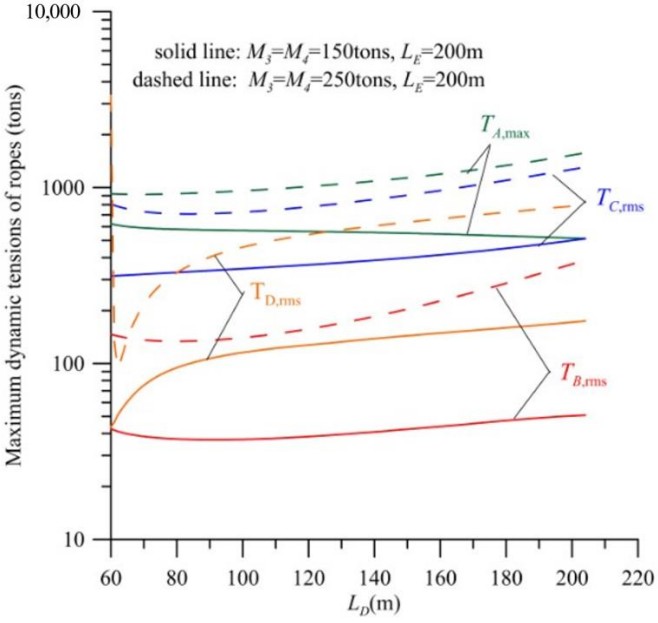

**Figure 9.** Dynamic tension of the four ropes under the typhoon irregular wave as a function of the diving length $L_D$ and the mass of pontoons $M_3$ and $M_4$ for $L_E$ = 200 m.

On the eastern coast of Taiwan, the average velocity of the Kuroshio at a depth of 150 m is 0.65 m/s, and that at a depth of 30 m is 1.1 m/s [35]. It is well known that the potential energy of ocean current can be estimated by using the formula $\eta \frac{1}{2}\rho A V^3$, where $h$ is the efficiency, $r$ is the density, $A$ is the operating area and $V$ is the flow velocity. Based on the formula, the ratio of the potential power generation of the diving depth of the turbine $L_D$ = 30 m to that of $L_D$ = 150 m is about 4.85. In other words, the deeper the diving depth of the turbine $L_D$, the smaller the power generation.

Figure 10a demonstrates the dynamic displacements of the turbine, floating platform and pontoons.

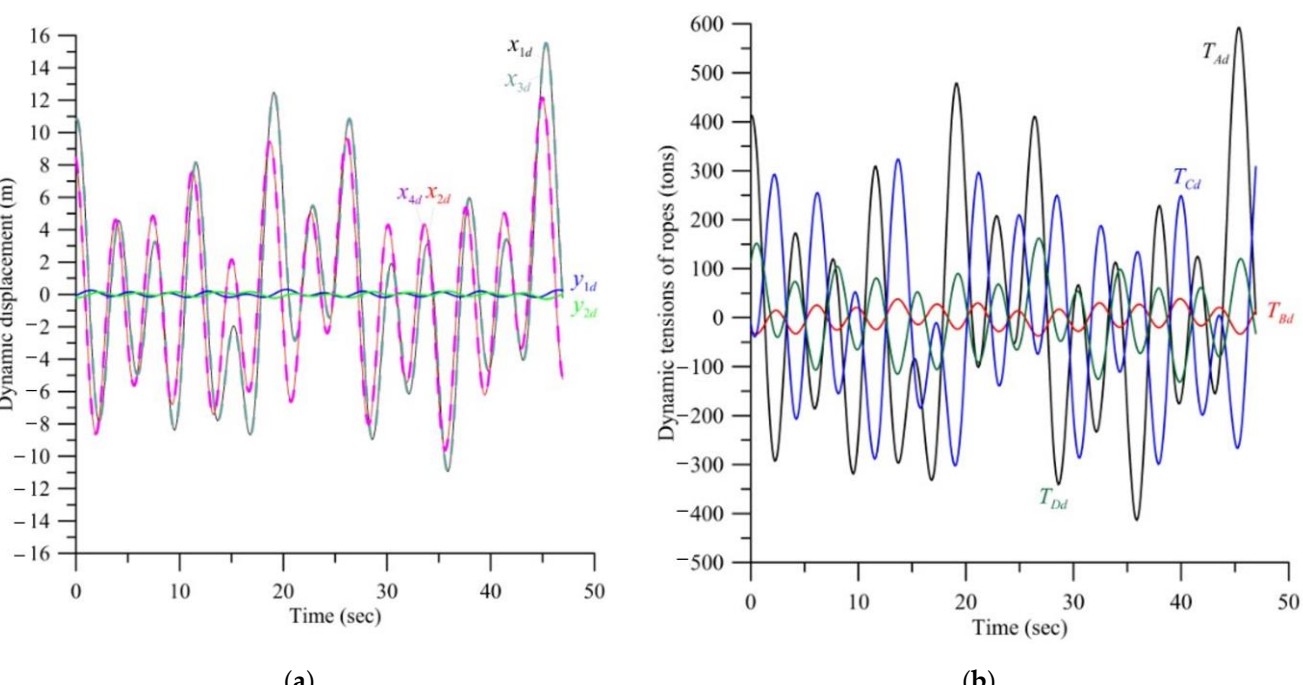

(**a**)                                                  (**b**)

**Figure 10.** (**a**) Dynamic displacements of the four elements and (**b**) dynamic tensions of ropes under typhoon irregular wave for $L_C = 60$ m, $L_D = 70$ m.

The diving depth of the floating platform $L_C = 60$ m, the diving depth of the turbine $L_D = 70$ m and the mass of pontoons $M_3 = M_4 = 150$ tons. The other parameters are the same as those in Figure 9. Dynamic displacements are multi-frequency coupled. The horizontal displacements of the turbine and the floating platform $y_{1d}$ and $y_{2d}$ are very small, the amplitude is about 0.30 m, the vertical displacements $x_{1d}$ and $x_{3d}$ are large and the amplitude is about 15.5 m, which is close to the significant wave $H_S = 15.4$ m. The amplitudes of vertical displacements $x_{2d}$ and $x_{4d}$ are about 15.5 m. The amplitudes of vertical displacements $x_{1d}$ and $x_{3d}$ are about 9.69 m. The vertical displacements of pontoon 3 and the floating platform directly connected by using rope C are synchronized and similar. The vertical displacements of pontoon 4 and the turbine directly connected by using rope D are synchronized and similar.

Figure 10b demonstrates the dynamic tension of the rope. The maximum dynamic tension $T_{A,max}$ of rope A connecting the floating platform and the mooring foundation is about 589 tons. The maximum dynamic tension $T_{B,max}$ of rope B connecting the turbine and the floating platform is about 38 tons. The maximum dynamic tension $T_{C,max}$ of rope C connecting pontoon 3 and the floating platform is about 322 tons. The maximum dynamic tension $T_{D,max}$ of rope D connecting pontoon 4 and the turbine is about 75 tons.

Figure 11a demonstrates the dynamic displacements of the turbine, floating platform and pontoons. The diving depth of the floating platform $L_C = 150$ m, the diving depth of the turbine $L_D = 60$ m. The other parameters are the same as those in Figure 6. Dynamic displacements are multi-frequency coupled. The horizontal displacements of the turbine and the floating platform $y_{1d}$ and $y_{2d}$ are very small, the amplitude is about 0.14 m, the amplitudes of vertical displacements $x_{1d}$ and $x_{3d}$ are about 8.6 m and the amplitudes of vertical displacements $x_{2d}$ and $x_{4d}$ are about 9.6 m, which are significantly lower than the significant wave $H_S = 15.4$ m. The vertical displacements of pontoon 3 and the floating platform directly connected by using rope C are synchronized and similar. The vertical displacements of pontoon 4 and the turbine directly connected by using rope D are synchronized and similar.

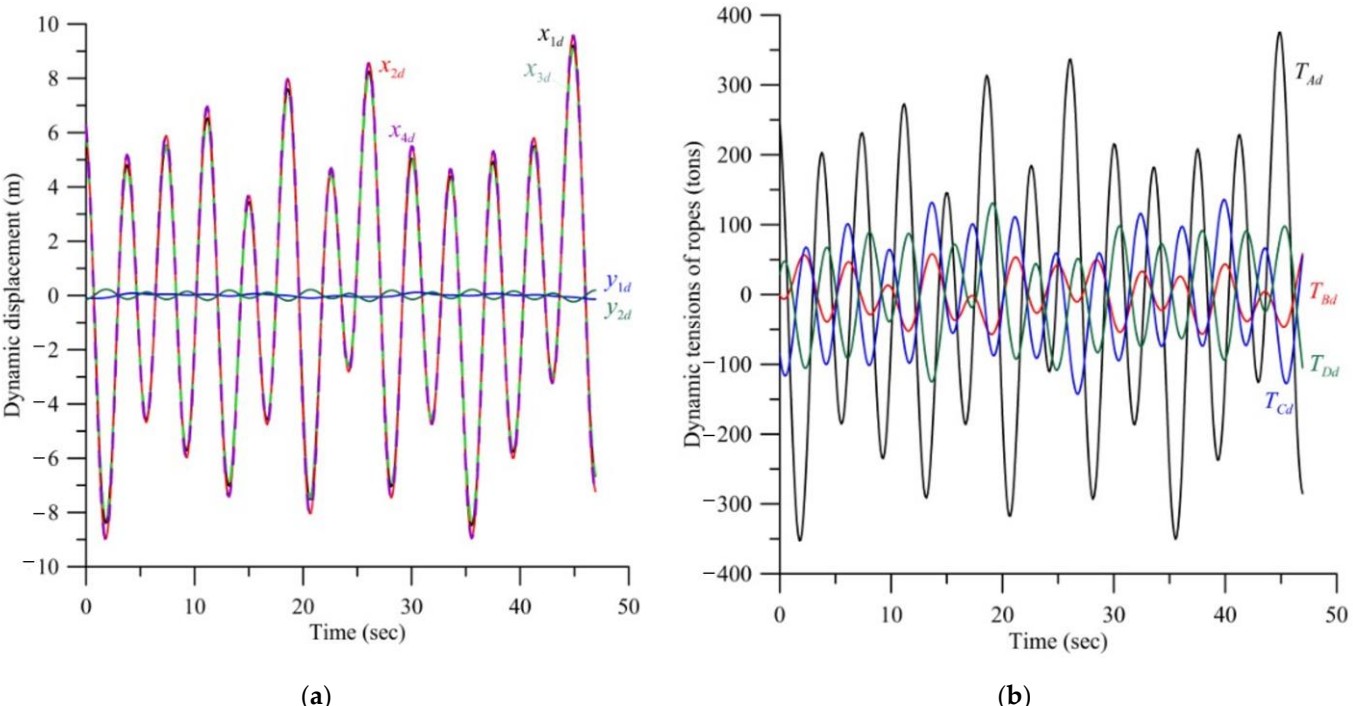

**Figure 11.** (**a**) Dynamic displacements of the four elements and (**b**) dynamic tensions of ropes under typhoon irregular wave for $L_C = 150$ m, $L_D = 60$ m.

Figure 11b demonstrates the dynamic tension of rope. The maximum dynamic tension $T_{A,max}$ of rope A connecting the floating platform and the mooring foundation is about 375 tons. The maximum dynamic tension $T_{B,max}$ of rope B connecting the turbine and the floating platform is about 58 tons. The maximum dynamic tension $T_{C,max}$ of rope C connecting pontoon 3 and the floating platform is about 143 tons. The maximum dynamic tension $T_{D,max}$ of rope D connecting pontoon 4 and the turbine is about 131 tons.

Figure 12 demonstrates the effects of the diving depth of the floating platform $L_C$ and the buffer spring connected in series with ropes C and D on the dynamic tension of the rope. The diving depth of the turbine $L_D = 60$ m. The effective spring constants of the two buffer springs are $K_{C,spring} = K_{D,spring} = K_{\text{rope A}}$. The other parameters are the same as those in Figure 5. Compared with Figure 5, it is found that the dynamic tensions $T_{A,max}$, $T_{B,max}$ and $T_{C,max}$ of the ropes A, B and C are significantly reduced at the resonance point, but the effect on $T_{D,max}$ is not obvious and is still over the fracture strength $T_{fracture}$. If the diving depth of the floating platform $L_C > 72$ m, the effect of the buffer springs on the dynamic tensions is negligible. It is concluded that the effect of the buffer springs on the dynamic tensions of this mooring system is slight.

Figure 13 demonstrates the effects of the cross-sectional area of pontoon $A_{BX}$, $A_{PX}$ and the diving depth of the floating platform $L_C$ on the dynamic tensions of the four ropes. The cross-sectional area of the two pontoons is $A_{BX} = A_{PX} = 4$ m$^2$. The other parameters are the same as those in Figure 5. Compared with Figure 5, it is found that the dynamic tensions are significantly increased. At the resonance point, the dynamic tension is over the fracture strength $T_{fracture}$. If the diving depth of the floating platform $L_C > 85$ m, the dynamic tension is close to the fracture strength $T_{fracture}$. It is concluded that the larger the cross-sectional area of the pontoon, the larger the dynamic tension.

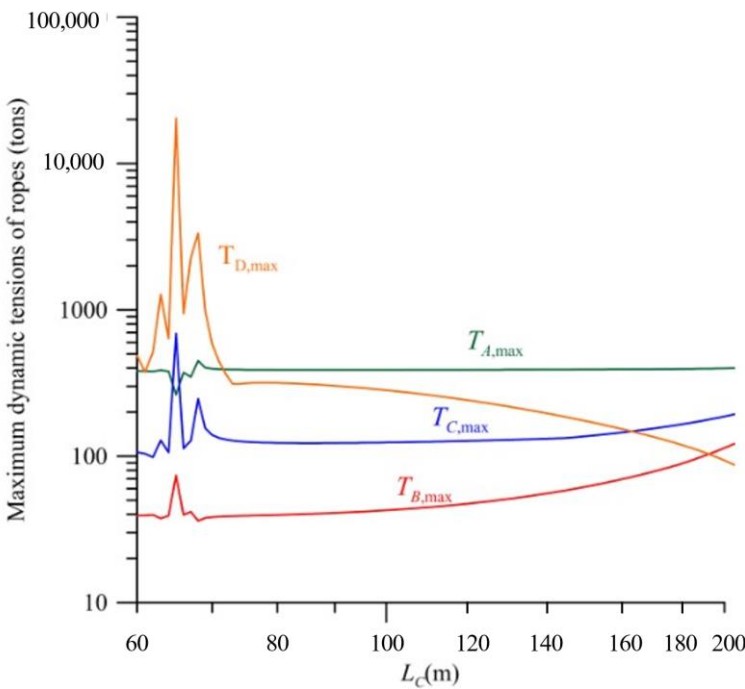

**Figure 12.** Dynamic tension of the four ropes under the typhoon irregular wave as a function of the diving length $L_C$ and the buffer springs $K_{C,spring}$, $K_{D,spring}$.

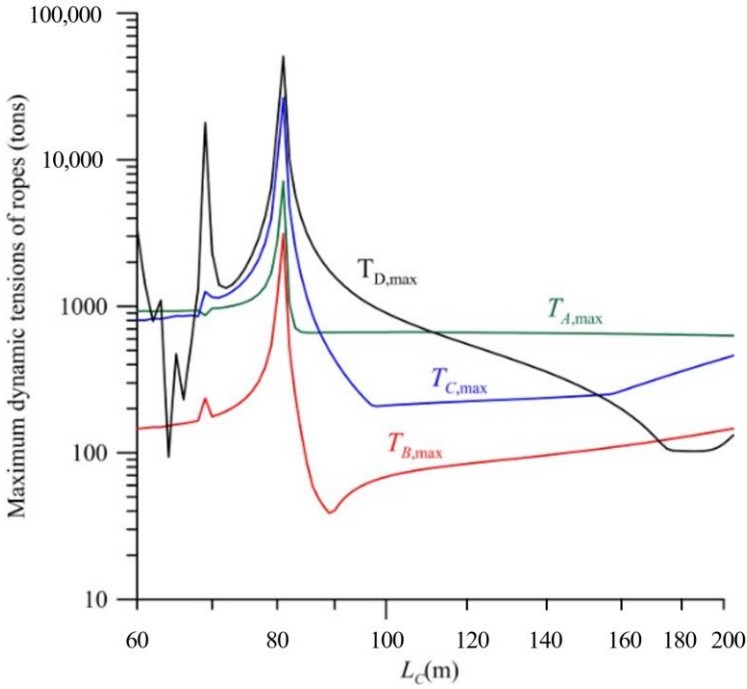

**Figure 13.** Dynamic tension of the four ropes under the typhoon irregular wave as a function of the cross-sectional area of the two pontoons $A_{BX}$, $A_{PX}$ and the diving depth of the floating platform $L_C$.

Figure 14 demonstrates the effects of the significant wave height $H_s$ and the peak period $P_w$ on the dynamic tension $T_{A,max}$. Based on Equations (14)–(16), the irregular wave is simulated by six regular waves, i.e., $n = 6$. The six regular waves share according to the ratio of energy {2,35,8,4,3,1}. The amplitude $a_i$, frequency $f_i$, the wave number $k_i$ and wave length $l_i$ can be determined. The diving depths $L_C = 60$ m and $L_D = 150$ m. The horizontal distance between the turbine and floating platform $L_E = 200$ m. Two buffer springs are connected in series with ropes C and D. The effective spring constants of the

two buffer springs are $K_{C,spring} = K_{D,spring} = K_{rope\ A}$. The other parameters are the same as those of Figure 5. It is found that the more the significant wave height $H_s$, the larger the dynamic tension $T_{A,max}$. For the peak period $T_p = 13.5$ s, the dynamic tension $T_{A,max}$ increases dramatically with the significant wave height Hs. With the increase of the peak period $T_p$, the increase rate of the dynamic tension $T_{A,max}$ becomes low.

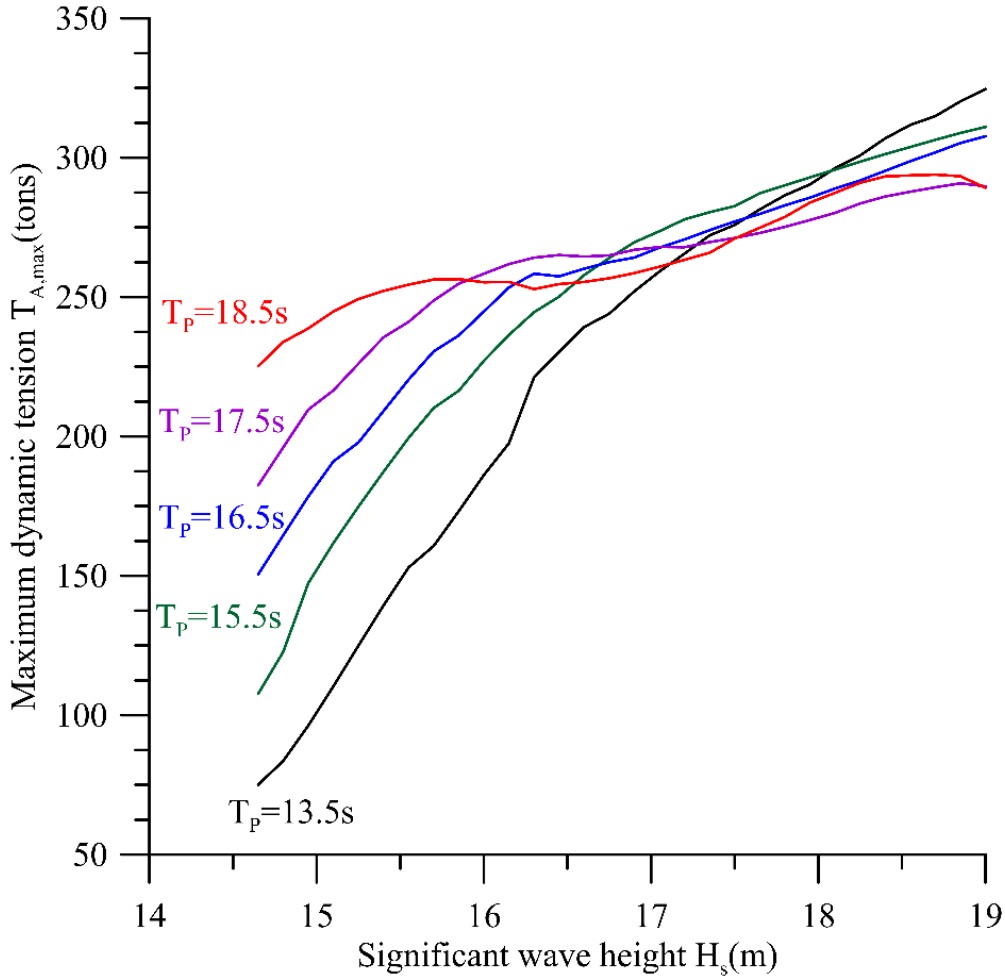

**Figure 14.** Dynamic tension $T_{A,max}$ under the typhoon irregular wave as a function of the significant wave height $H_s$ and the peak period $T_p$.

Figure 15 demonstrates the effect of the relative angle a of the wave and ocean current on the dynamic tension. The significant wave height $H_s = 15$ m, and the peak period $P_w = 16.5$ s. The other parameters are the same as those of Figure 14. It is observed that the effect of the relative angle a of the wave and ocean current on the dynamic tension is significant. $T_{A,max}(\alpha = 180°)$ and $T_{C,max}(\alpha = 180°)$ are much larger than $T_{A,max}(\alpha = 0°)$ and $T_{C,max}(\alpha = 0°)$, respectively. Moreover, $T_{D,max}(\alpha = 90°)$ is significantly larger than $T_{A,max}(\alpha = 0°)$ and $T_{A,max}(\alpha = 180°)$.

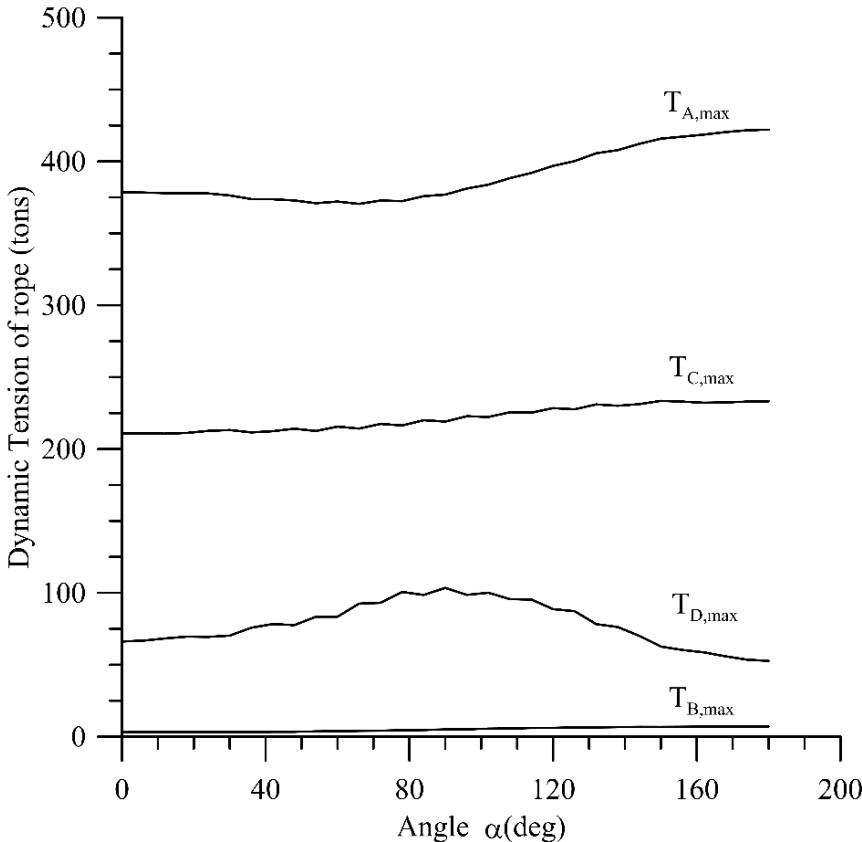

**Figure 15.** Dynamic tension of the four ropes under the typhoon irregular wave as a function of the relative angle $\alpha$.

Based on the frequency Equation (62), the effects of the masses $M_3$, $M_4$ and $M_2$ and the distance $L_E$ and the areas on the natural frequencies are investigated and listed in Table 2. It is found that the larger the cross-sectional areas of pontoon $A_{Bx}$ and $A_{BT}$, the higher the natural frequencies of the system. The larger the masses of pontoon $M_3$ and $M_4$, the lower the natural frequencies of the system. The larger the mass of turbine $M_2$, the lower the first natural frequency of the system. However, the effect of the mass of turbine $M_2$ on the second natural frequency of the system is negligible. The larger the distance between the turbine and the floating platform $L_E$, the higher the second natural frequency of the system. However, the effect of the distance $L_E$ on the first natural frequency of the system is negligible.

**Table 2.** The first two natural frequencies $f_{n1}$ and $f_{n2}$ as a function of the masses $M_3$, $M_4$ and $M_2$, the distance $L_E$ and the areas $A_{Bx}$, $A_{BT}$ for $M_1 = 300$ tons.

| $L_E$ (m) | $M_3$, $M_4$ (tons) | $M_2$ (tons) | $A_{Bx} = A_{BT} = 2.12$ m$^2$ | | $A_{Bx} = A_{BT} = 4$ m$^2$ | |
|---|---|---|---|---|---|---|
| | | | $f_{n1}$ (Hz) | $f_{n2}$ (Hz) | $f_{n1}$ (Hz) | $f_{n2}$ (Hz) |
| 130 | | 838 | 0.0220 | 0.0703 | 0.0302 | 0.0761 |
| | | 535 | 0.0258 | 0.0703 | 0.0355 | 0.0761 |
| | 250 | | | | | |
| 200 | | 838 | 0.0220 | 0.0806 | 0.0302 | 0.0857 |
| | | 535 | 0.0258 | 0.0806 | 0.0355 | 0.0857 |
| 130 | 150 | 535 | 0.0277 | 0.0776 | 0.0379 | 0.0839 |
| 200 | | | 0.0277 | 0.0890 | 0.0380 | 0.0946 |

## 4. Conclusions

This paper studies the safe design of a mooring system for an ocean current generator that is working under the impact of typhoon waves. Two mooring designs are investigated, and one safe and feasible mooring system is proposed. The proposed mooring design can stabilize the turbine and platform around a certain predetermined water depth, thereby, maintaining the stability and safety of the ocean current generator. The effects of several parameters on the dynamic response under irregular wave impact were discovered as follows:

(1) Considering the first mooring configuration, the diving depth $L_D$ of the turbine is fixed at 60 m. When the diving depth of the floating platform $L_C$ =150 m, the dynamic tension is significantly less than the fracture strength $T_{Fracture}$ of rope, and it is far from the resonance. Moreover, because the diving depth $L_D$ of the turbine is far from the depth $L_C$ of the floating platform, the floating platform does not interrupt the turbine water flow. Because the floating platform is a structure without a rotating mechanism in it, such as the rotating blade of a turbine, the water-proof at the depth of 150 m under sea surface is easily constructed. Therefore, this mooring configuration is safe and feasible;

(2) Considering the second mooring configuration, the diving depth $L_C$ of the floating platform is fixed at 60 m. When the diving depth $L_D$ of the floating platform is larger than the diving depth $L_C$, there is no resonance point, but the dynamic tension $T_{Ad,max}$ of rope A is obviously larger than that of the first method and close to the fracture strength $T_{Fracture}$. It is found [32] that, for the Kuroshio current on the eastern coast of Taiwan, the greater the depth under the sea surface, the lower the current flow rate. The ratio of the potential power generation of the diving depth of the turbine $L_D$ = 30 m to that of $L_D$ = 150 m is about 4.85. Moreover, because there are the rotating blades of the turbine, the water-proof at the higher pressure under sea surface is difficult to construct. Therefore, the second mooring configuration is not recommended;

(3) The larger the area of pontoons $A_{BX}$ and $A_{TX}$, the larger the maximum dynamic tensions, especially for $T_{Ad,max}$;

(4) For the first mooring configuration, if the weight of the pontoon is too low, the dynamic displacement of the system is too intense, resulting in the excessive dynamic tension of the rope;

(5) The effect of the buffer springs on the dynamic tensions of the first mooring configuration is slight.

The coupled translational and rotational motions will be studied in another manuscript. Moreover, the transient response of the system subjected to impact force will be investigated in the future.

**Author Contributions:** Conceptualization, S.-M.L. and C.-T.L.; methodology, S.-M.L.; software, S.-M.L. and D.-W.U.; validation, S.-M.L.; formal analysis, S.-M.L.; investigation, S.-M.L. and D.-W.U.; resources, S.-M.L. and C.-T.L.; data curation, D.-W.U.; writing—original draft preparation, S.-M.L.; writing—review and editing, C.-T.L.; visualization, S.-M.L.; supervision, S.-M.L.; funding acquisition, S.-M.L. and C.-T.L. All authors have read and agreed to the published version of the manuscript.

**Funding:** This work was financially supported by the Green Energy Technology Research Center from The Featured Areas Research Center Program within the framework of the Higher Education Sprout Project by the Ministry of Education (MOE) in Taiwan and the National Academy of Marine Research of Taiwan, R. O. C. (NAMR110051).

**Institutional Review Board Statement:** Not applicable.

**Informed Consent Statement:** Not applicable.

**Data Availability Statement:** The figures and the tables in this manuscript have clearly described all the data of this study.

**Acknowledgments:** This work was financially supported by the Green Energy Technology Research Center from The Featured Areas Research Center Program within the framework of the Higher Education Sprout Project by the Ministry of Education (MOE) in Taiwan and the National Academy of Marine Research of Taiwan (NAMR110051).

**Conflicts of Interest:** The authors declare no conflict of interest. The funders had no role in the design of the study; in the collection, analyses, or interpretation of data; in the writing of the manuscript, or in the decision to publish the results.

## Nomenclature

| | |
|---|---|
| $A_{BX}$, $A_{BT}$ | cross-sectional area of pontoons 3 and 4, respectively |
| $A_{BY}$, $A_{TY}$ | damping area of platform and turbine under current, respectively |
| $a_i$ | amplitude of the *i*-th regular wave |
| $C_{DFy}$, $C_{DTy}$ | damping coefficient of floating platform and turbine |
| $F_B$ | buoyance |
| $F_D$ | drag under current |
| $f$ | wave frequency |
| $H_{bed}$ | depth of seabed |
| $H_s$ | significant wave height |
| $g$ | gravity |
| $K$ | effective spring constant |
| $\overrightarrow{K}_i$ | wave vector of the *i*-th regular wave |
| $\widetilde{k}_i$ | wave number of the *i*-th regular wave |
| $L_i$, $i = $ A,B,C,D | length of rope i |
| $L_i$ | length of rope i |
| $M_i$ | mass of element i |
| $m_{eff,x}$, $m_{eff,y}$ | vertical and horizontal effective mass of rope A, respectively |
| $P_w$ | peak period of wave |
| $\overrightarrow{R}$ | coordinate |
| $T_i$ | tension force of rope i |
| $t$ | time variable |
| $V$ | ocean current velocity |
| $x_i$, $i = 1{\sim}4$ | vertical displacements of the floating platform, the turbine and the pontoons, respectively |
| $x_w$ | sea surface elevation |
| $y_1$, $y_2$ | horizontal displacements of the floating platform and the turbine, respectively |
| $\alpha$ | relative angle between the directions of wave and current |
| $\rho$ | density of sea water |
| $\Omega_i$ | angular frequency of the *i*-th regular wave |
| $\omega$ | angular frequency |
| $\varphi_i$ | phase angle of the *i*-th regular wave |
| $\phi_i$ | phase delay of the *i*-th regular wave |
| $\theta_i$ | angles of rope i |
| $\lambda_i$ | length of the *i*-th regular wave |
| $\delta_i$ | elongation of rope i |

## Subscript

| | |
|---|---|
| 0~4 | mooring foundation, floating platform, turbine and two pontoons, respectively |
| A, B, C, D | Ropes A, B, C and D, respectively |
| s, d | static and dynamic, respectively |
| PE | high-strength PE dyneema rope |
| p | peak |

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
