# Peer review of "Design and Dynamic Stability Analysis of a Submersible Ocean Current Generator Platform Mooring System under Typhoon Irregular Wave"

_jmse, doi:10.3390/jmse10040538_

Round 1

Reviewer 1 Report

The authors studied the engineering design of a mooring system to extract energy from ocean currents east of Taiwan. This cause was definitely worthwhile. However, the style of presentation and scientific explanations needed improvements before the paper could be accepted.

(1) The irregular wave was represented as six waves. There did not appear to be an adequate explanation why six waves would be appropriate. Were the , wavelengths of these six representatives of typhoon situation?

(2) There were too many parameters in the formulation. How could we be sure that the appropriate values were chosen?

(3) The analysis presented was too technical and too involved for people outside of this field to follow. The symbols were confusing too.  Presentations should be condensed.

(4) Was there any simple physical explanation on the occurrence of local maxima in the tension force, e.g. those in Figure 7?

Author Response

Dear Editor Mastej:

I have revised the paper (JMSE-1616071-R1) according to your directions and the suggestions of the reviewers and resubmit it for publication in JMSE. The changes are marked in red color. I would like to thank the reviewers for their excellent job and JMSE for considering my manuscript for publication.

Sincerely yours,

Shueei-Muh Lin

Professor

(E-mail: smlin45@gmail.com)

Reviewer 2 Report

Please see attached word file

Author Response

(The authors gave the same response as above.)

Round 2

Reviewer 1 Report

(1) There are still way too many symbols in the calculations, more than any reasonably trained scientist can absorb. Perhaps a list of symbols at the end of the paper will be helpful.

(2) Some curves in the Figures seem to have 'cusps' / sharp corners. Perhaps the authors can refine the resolution or check the numerical results there.

Author Response

Dear Editor Mastej and reviewers:

I have re-revised the paper (JMSE-1616071-R2) according to your directions and the suggestions of the reviewers and resubmit it for publication in JMSE. The changes are marked in red color. I would like to thank the reviewers for their excellent job and JMSE for considering my manuscript for publication.

Sincerely yours,

Shueei-Muh Lin

Professor

(E-mail: smlin45@gmail.com)

Reviewer 2 Report

Revised Manuscript: “Design and Dynamic Stability Analysis of a Submersible Ocean Current Generator-Platform Mooring System Under Typhoon Irregular Wave

The authors have addressed the majority of the comments of the reviewer by (i) adding clarifications, (ii) providing more information (e.g. run an extra cases for different Hs, Tp and wave relative angle), (iv) generating new figures and tables, and (v) updating the introduction and conclusions, as suggested by the reviewer. These updates and new material shed light on the underlying physics of the presented results and help explain the observed trends better.

Overall, the aforementioned changes increased the size of the manuscript from 28 to 32 pages and improved its quality, increasing its value for the reader. This is a testament of the authors’ sincere effort to address diligently the comments of the reviewers. So, thank you for your effort.

There are only a few remaining comments, as shown below, which the authors can address easily. Therefore, the manuscript is suggested to be accepted for publication with minor revisions.

Comments:

Authors reply to comment 5: The authors mention some interesting facts in their reply as well as some limitations of the current manuscript (e.g. it focuses only on the translational motion, not the pitching, rolling, yawing).  For clarity purposes, the authors are advised to add some of these comments in the actual manuscript.

Authors reply to comment 8: The authors mention that the dynamic response of structure under wave impulsive forces is a topic that will be discussed in future research. That’s ok. Please mention in the actual manuscript that the effect of impulsive wave forces will be investigated in future research, and explain to the reader that such effects could be potentially significant as documented in the reference mentioned in the original comment ([n9]).

Authors reply to comment 9: The authors provide some interesting arguments here, so it would be useful for the reader, if they could include some of these arguments in the actual text. Moreover, the authors mention that breaking/broken waves and bores are not investigated in the current manuscript and will be addressed in future research. That's fine. Please mention this limitation it in the manuscript, and also state the potential differences/complexities of breaking/broken waves as seen in previous studies (e.g. [n13, n14] of the original comment) so that the reader can understand why future investigation of such waves types in important.

Authors reply to comment 10: The last bullet point is going to be quite helpful for the reader, so please include it in the actual manuscript.

Author Response

(The authors gave the same response as above.)
